# Beyond Fine-Tuning: A Systematic Study of Sampling Techniques in Personalized Image Generation

## Abstract

Personalized text-to-image generation focuses on creating customized images based on user-defined concepts and text descriptions. A good balance between learned concept fidelity and its ability to be generated in different contexts is a major challenge in this task. Modern personalization techniques often strive to find this balance through diverse fine-tuning parameterizations and enhanced sampling methods that integrate superclass trajectories into the backward diffusion process. Improved sampling methods present a cost-effective, training-free way to enhance already fine-tuned models. However, outside of fine-tuning approaches, there is no systematic analysis of sampling methods in the personalised generation literature. Most sampling techniques are introduced alongside fixed fine-tuning parameterizations, which makes it difficult to identify the impact of sampling on the generation outcomes and whether it can be applied with other fine-tuning strategies. Moreover, they don't compare with the naive sampling approaches, so the intuition of how the superclass trajectory affects the sampling process remains underexplored. In this work, we propose a systematic and comprehensive analysis of personalized generation sampling strategies beyond the fine-tuning methods. We explore various combinations of concept and superclass trajectories, developing a deep understanding of how superclass influence generation outputs. Based on these results, we demonstrate that even a weighted mix of the concept and superclass trajectory can establish a strong baseline that enhances the adaptability of concepts across different contexts and can be effectively transferred to any training strategy, including various fine-tuning parameterizations, text embedding optimization, and hypernetworks. We analyze all methods through the lens of the trade-off between concept fidelity, editability, and computational efficiency, ultimately providing a framework to determine which sampling method is most suitable for specific scenarios.

## 1 Introduction

Diffusion-based text-to-image generation models (Ramesh et al., 2022; Saharia et al., 2022; Rombach et al., 2022a), trained on large datasets, have recently achieved impressive results in generating photorealistic images from textual prompts. Despite their advanced performance, these models are limited when it comes to generating user-defined concepts, which are difficult to describe accurately with text alone. This limitation has led to a growing interest in the field of subject-driven text-to-image generation (Ruiz et al., 2023; Gal et al., 2022). In this task, given a small image dataset (3-5 images) of a given subject, we want to introduce the knowledge of this subject into the pre-trained text-to-image diffusion model and learn to generate it in different contexts described by textual prompts.

Simultaneously preserving the identity of the concept and the ability to adapt it to the new context is a difficult balance to achieve and the main challenge in personalized image generation. On the one hand, the model must generate high-fidelity images of the concepts,

even if it has never encountered them during the pre-training phase. On the other hand, the model should not overfit in order to retain the ability to follow different textual descriptions of the scenes.

To achieve a better balance between concept fidelity and editability, modern methods introduce a variety of training process improvements. These include fine-tuning parameterizations (Ruiz et al., 2023; Gal et al., 2022; Kumari et al., 2023; Han et al., 2023; Tewel et al., 2023; Qiu et al., 2024), regularizations (Ruiz et al., 2023; Kumari et al., 2023), and encoder-based paradigms (Wei et al., 2023). For more detailed review see Appendix A. Another direction is to utilize sampling methods applied after training to enhance an already fine-tuned model. The main idea of such methods (Zhou et al., 2023; Gu et al., 2024) is to combine the sampling trajectories of prompts with concept and superclass tokens (e.g., for a dog concept we mix trajectories for two prompts: *"a purple V*"* and *"a purple dog"*, see Figure 1). The sampling-based approaches can provide a cost-effective, training-free way to improve the balance between concept identity and its editability. While fine-tuning and sampling methods are two distinct strategies to addressing the same issue, current research often does not distinguish between these methodologies. As an example, current works (Zhou et al., 2023; Gu et al., 2024) introduce complex sampling procedures alongside fixed fine-tuning, leaving unclear the impact of sampling on generation results and whether it can be integrated with other fine-tuning strategies. Furthermore, they do not compare the proposed strategies against naive sampling approaches, resulting in a lack of insight into how superclass trajectories influence the sampling process. In summary, the personalized generation sampling process remains underexplored, with three main open challenges: (1) *The impact of superclass trajectory integration is under-researched*, as previous work has not fully elucidated how the incorporation of superclass trajectories affects the generation output. (2) *Simple sampling baselines are often overlooked*, and their potential remains undervalued. (3) *Limitations imposed by fine-tuning strategies*; current sampling methods are almost always tied to specific fine-tuning schemes, which restricts the ability to study sampling independently and hampers fair comparisons between different approaches.

To address these challenges, we propose several contributions aimed at advancing the understanding and application of sampling strategies in personalized text-to-image generation. Our work explores the impact of sampling methods beyond fine-tuning strategies, establishing simple yet powerful baselines. Specifically, we make the following key contributions:

**1. A systematic and comprehensive analysis of how superclass trajectories influence the sampling process.** We investigate various combinations of concept and superclass trajectories, including switching, mixed, and masked sampling techniques, along with their hybrid variants. We carefully ablate hyperparameters across all methods, assess their importance, and retain only the most impactful ones.

**2. A finetuning-independent evaluation of various sampling strategies.** We compare various sampling methods, including naive approaches, applied to a fixed fine-tuned model to analyze the impact of the sampling beyond the fine-tuning strategy. Moreover, we demonstrate how these strategies can be applied effectively across different fine-tuning methods, including various fine-tuning parameterizations, text embedding optimization, and hypernetworks.

**3. A framework for selecting the most suitable sampling method for specific generation tasks.** We perform a fair comparison of sampling methods based on trade-offs between concept fidelity, adaptability, and computational efficiency and build a framework for determining the most appropriate sampling method for specific scenarios.

## 2 PRELIMINARIES

**Stable Diffusion Model** As a base model in this work, we utilize Stable Diffusion (Rombach et al., 2022b), one of the most widely used diffusion model in research. Stable Diffusion is a large text-to-image model that is trained on pairs $(x, P)$, where $x$ is an image and $P$ is a text prompt describing it. Stable Diffusion includes the CLIP (Radford

et al., 2021) text encoder $E_T$, which is used to obtain the text conditional embedding $p = E_T(P)$, the encoder $E$, which transforms the input image into the latent space $z = E(x)$, the decoder $D$, which reconstructs the input image from the latent $x \approx D(z)$, and a UNet-based (Ronneberger et al., 2015) conditional diffusion model $\varepsilon_\theta$. The denoising process is performed in the latent space. With a randomly sampled noise $\varepsilon \sim N(0, I)$, the time step t and the coefficients controlling the noise schedule we obtain a noisy latent code: $z_t = \alpha_t z + \sigma_t \varepsilon$. The goal of UNet $\varepsilon_\theta$ is to predict the noise from the noisy latent:

$$\min_\theta \mathbb{E}_{p,z,\varepsilon,t} \left[ \| \varepsilon - \varepsilon_\theta(z_t, p) \|_2^2 \right] \tag{1}$$

During inference, a random noise $z_T \sim N(0, I)$ is denoised step by step to $z_0$, using DDIM sampling Song et al. (2020): $z_{t-1} = \text{DDIM}(t, z_t, \varepsilon_\theta(z_t, p))$, $t = T, \ldots, 1$. The resulting image is obtained through the decoder as $D(z_0)$.

**Classifier-free guidance** A commonly used technique to improve the generation quality of conditional diffusion models post-training is classifier-free sampling (Ho & Salimans, 2022). Given the current noisy sample $z_t$ and condition $p$, the diffusion model outputs the predictions of the conditional noise $\varepsilon_\theta(z_t, E_T(p))$ and unconditional noise $\varepsilon_\theta(z_t)$ (conditioned on null text). Then an updated prediction

$$\tilde{\varepsilon}_\theta(z_t, p) = \varepsilon_\theta(z_t) + \omega(\varepsilon_\theta(z_t, p) - \varepsilon_\theta(z_t)) \tag{2}$$

will be used to sample $z_{t-1}$, where $\omega$ is a guidance scale.

**Finetuning for Personalized Text-to-Image Generation** Let $\mathbb{C} = \{x\}_{i=1}^N$ be a small image set of images with a specific concept. A special text token $V^*$ can be bind to it, using the following fine-tuning objective:

$$\min_\theta \mathbb{E}_{z=\mathcal{E}(x), x \in \mathbb{C}, \varepsilon, t} \left[ \| \varepsilon - \varepsilon_\theta(z_t, p^C) \|_2^2 \right] \tag{3}$$

where $p^C = E_T(P^C)$ is a text embedding of the prompt $P^C = $ "a photo of a V*"

## 3 METHODS

Given a model $\varepsilon_\theta$, already fine-tuned by (3) for a specific concept, we can identify two distinct sampling approaches, each maximizing one of the objectives: concept fidelity or editability:

Sampling with concept: $\quad \tilde{\varepsilon}_\theta(z_t, p^C) = \varepsilon_\theta(z_t) + \omega(\varepsilon_\theta(z_t, p^C) - \varepsilon_\theta(z_t)) \quad$ (4)

Sampling with superclass: $\quad \tilde{\varepsilon}_\theta(z_t, p^S) = \varepsilon_\theta(z_t) + \omega(\varepsilon_\theta(z_t, p^S) - \varepsilon_\theta(z_t)) \quad$ (5)

Here, $p^C$ represents a concept prompt embedding (for example, *"a V\* with a city in the background"*) and $p^S$ indicates a superclass prompt embedding (*"a backpack with a city in the background"*) where the concept token $V^*$ is replaced by a superclass token (*"backpack"*).

The extended fine-tuning of the model $\varepsilon_\theta$ enhances its ability to accurately reproduce the concept generated via (4). However, this improvement comes at the cost of overlooking the contextual information supplied by the prompt $P^C$ (see Figure 1a). Conversely, the generation via (5) ensures the highest alignment with the text prompt, though at the expense of preserving the concept's identity (see Figure 1b).

This raises the question of whether we can integrate the two sampling strategies (4) and (5) to obtain the optimal balance between the high fidelity of the learned concept identity and its adaptability to various contexts.

### 3.1 MIXED SAMPLING

One reasonable approach for incorporating superclass into the generation process (Zhou et al., 2023) is to modify the sampling strategy by adding guidance to the superclass prompt (see Figure 1c):

$$\tilde{\varepsilon}_\theta^{MX}(z_t, p^S, p^C) = \varepsilon_\theta(z_t) + \omega_s(\varepsilon_\theta(z_t, p^S) - \varepsilon_\theta(z_t)) + \omega_c(\varepsilon_\theta(z_t, p^C) - \varepsilon_\theta(z_t)) \tag{6}$$

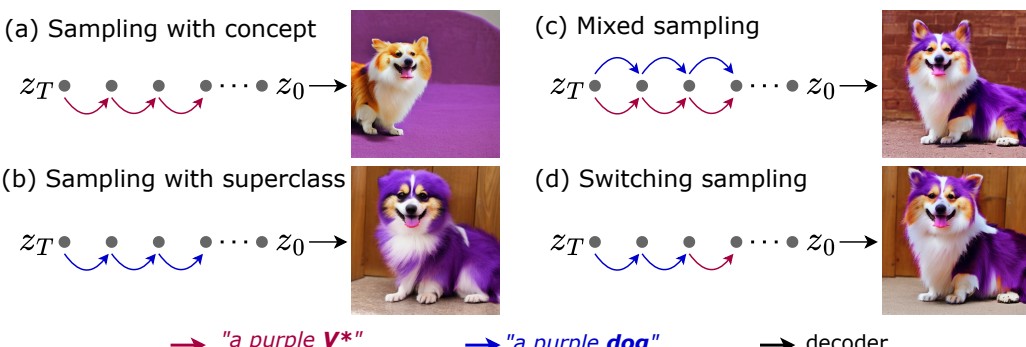

Figure 1: **Visualization of Different Sampling Strategies.** (a) Usual sampling with concept reproduces the concept but does not align closely with the text prompt. (b) Generation with superclass effectively captures the context derived from the prompt but produces a random superclass representative (e.g., dog). (c-d) Mixed and Switching sampling strategies enhance context preservation while maintaining the identity of the concept.

By adjusting the ratio between the concept guidance scale $\omega_c$ and the superclass guidance scale $\omega_s$, we can either amplify or diminish the influence of the concept or superclass, thus varying the trade-off between concept and context fidelity. In Figure 2, you can observe how the generated output alters with increasing superclass influence. For instance, in the teapot example, as we raise the superclass guidance scale, the context, which was initially poorly represented through sampling with the concept, gradually becomes more accurate. However, excessive superclass influence may result in a loss of concept identity preservation, as illustrated in the dog example.

### 3.2 Switching sampling

Another solution of how to combine the superclass sampling trajectory with the concept sampling trajectory is to condition several steps at the superclass prompt embedding $p^S$, then at the *switching step* $t_{sw}$ switch to the concept prompt embedding $p^C$ (see Figure 1d). In this case (2) will be rewritten in the following form

$$\tilde{\varepsilon}_\theta^{SW}(z_t, p^S, p^C, t_{sw}) = \varepsilon_\theta(z_t) + \begin{cases} \omega(\varepsilon_\theta(z_t, p^S) - \varepsilon_\theta(z_t)), & t > T - t_{sw} \\ \omega(\varepsilon_\theta(z_t, p^C) - \varepsilon_\theta(z_t)), & \text{otherwise} \end{cases} \quad (7)$$

By increasing the *switching step* $t_{sw}$, we can amplify the influence of the superclass and thus improve context preservation. Up to 10 steps can effectively recover context that has been poorly generated through standard sampling with concept, as demonstrated in the teapot example in Figure 2. Nonetheless, this strategy may result in notable degradation of the concept's identity. The effect of the superclass can be so intense that the concept loses its original attributes and takes on excessive characteristics from the superclass, as evidenced by the dog example in Figure 2.

This sampling procedure is similar to Photoswap Gu et al. (2024) approach adapted to the personalization task. The main difference is that in switched sampling we take the noise predictions entirely from the superclass trajectory for the first $t_{sw}$ steps, whereas in Photoswap only the self- and cross-attention maps and features are taken from the superclass for the first $t_{sw}$ steps. However, as we show in the Section 4, the results of these two methods are almost indistinguishable.

The aforementioned methods can be flexibly combined, we refer to this type of sampling as *multi-stage sampling*:

$$\tilde{\varepsilon}_\theta^{MS}(z_t, p^S, p^C) = \varepsilon_\theta(z_t) + \begin{cases} (\omega_s + \omega_c)(\varepsilon_\theta(z_t, p^S) - \varepsilon_\theta(z_t)) & t > T - t_{sw} \\ \omega_s(\varepsilon_\theta(z_t, p^S) - \varepsilon_\theta(z_t)) + \omega_c(\varepsilon_\theta(z_t, p^C) - \varepsilon_\theta(z_t)) & \text{otherwise} \end{cases}$$

$$(8)$$

This combination enables a greater influence of the superclass on the generated output and enhances alignment with the text prompt. However, it is important to consider that as the influence of the superclass increases, the more the concept's identity is lost.

### 3.3 MASKED SAMPLING

Sampling with a superclass prompt hinders the preservation of concept identity, whereas sampling with a concept prompt disrupts contextual adaptation. To address this challenge, restricting the image regions impacted by each sampling approach could be beneficial. This can be effectively achieved through masking.

Suppose at each diffusion step we could obtain a concept mask $M_t$, then we can utilize it in the mixed sampling. Specifically, we apply this mask to the concept trajectory, ensuring it only influences relevant regions:

$$\varepsilon_\theta^M(z_t, p^S, p^C) = \varepsilon_\theta(z_t) + \omega(\varepsilon_\theta(z_t, p^C) - \varepsilon_\theta(z_t)) \odot M_t + \omega(\varepsilon_\theta(z_t, p^S) - \varepsilon_\theta(z_t)) \odot \overline{M_t} \quad (9)$$

Moreover, to enhance the alignment between regions inside and outside the mask, and to gently amplify the influence of the superclass within the mask—especially in cases where prompts alter the object's appearance (like color or outfit)—we can apply mixed sampling within the mask:

$$\begin{aligned} \varepsilon_\theta^M(z_t, p^S, p^C) = \varepsilon_\theta(z_t) &+ \omega_c(\varepsilon_\theta(z_t, p^C) - \varepsilon_\theta(z_t)) \odot M_t \\ &+ \omega_c(\varepsilon_\theta(z_t, p^S) - \varepsilon_\theta(z_t)) \odot M_t \\ &+ (\omega_c + \omega_s)(\varepsilon_\theta(z_t, p^S) - \varepsilon_\theta(z_t)) \odot \overline{M_t} \end{aligned} \quad (10)$$

The generation process begins with mixed sampling for a limited number of steps, thereby enhancing the robustness of mask generation. Subsequently, we apply masked sampling as described in (10), using the concept mask $M_t(q)$. This mask is derived by averaging the cross-attention maps associated with the concept identifier token across all U-Net layers and binarizing it using a threshold determined by the quantile $q$.

$$\tilde{\varepsilon}_\theta^M(z_t, p^S, p^C) = \begin{cases} \tilde{\varepsilon}_\theta^{MX}(z_t, p^S, p^C, \omega_c^0, \omega_s^0), & t > T - t_{sw} \\ \varepsilon_\theta^M(z_t, p^S, p^C, \omega_c, \omega_s, q), & \text{otherwise}, \end{cases} \quad (11)$$

where $\varepsilon_\theta^M(z_t, p^S, p^C, \omega_c, \omega_s, q)$ is computed as in (10).

Equation 11 summarizes full masked sampling algorithm. Increasing the quantile $q$ reduces the area influenced by the concept, thereby expanding the region impacted by the superclass (see Appendix E) and enhancing the influence of the context, as illustrated in Figure 2.

### 3.4 OTHER APPROACHES

**ProFusion** The main contribution of the Profusion (Zhou et al., 2023) sampling method is a novel technique to enforce the concept preservation combined with Mixed Sampling. A sampling step in this approach consist of the following stages: (1) we predict $x_t \to \tilde{x}_{t-1}$ through the usual diffusion backward sampling process with concept (2) after that we make a forward diffusion step $\tilde{x}_{t-1} \to \tilde{x}_t$ (3) finally, we again make a backward step with the Mixed sampling $\tilde{x}_t \to x_{t-1}$. The first two steps define Fusion Step and have a special hyperparameter $r$ that controls its intensity(e.g. the influence on the result). In case $r = 0$ we get Mixed sampling.

**Photoswap** In these method author propose to replace self-attention features, cross-attention maps and self-attention maps in the concept trajectory with those from the superclass during several initial steps. Thus, the method has three hyperparameters: (1) $t_{SF}$ the number of initial steps during which the self-attention features are replaced, (2)$t_{CM}$ the same parameter for cross-attention maps, and (3) $t_{SM}$ for self-attention maps.

### 3.5 EVALUATION PROTOCOL FOR SAMPLING TECHNIQUES

The study of sampling methods involves several key steps.

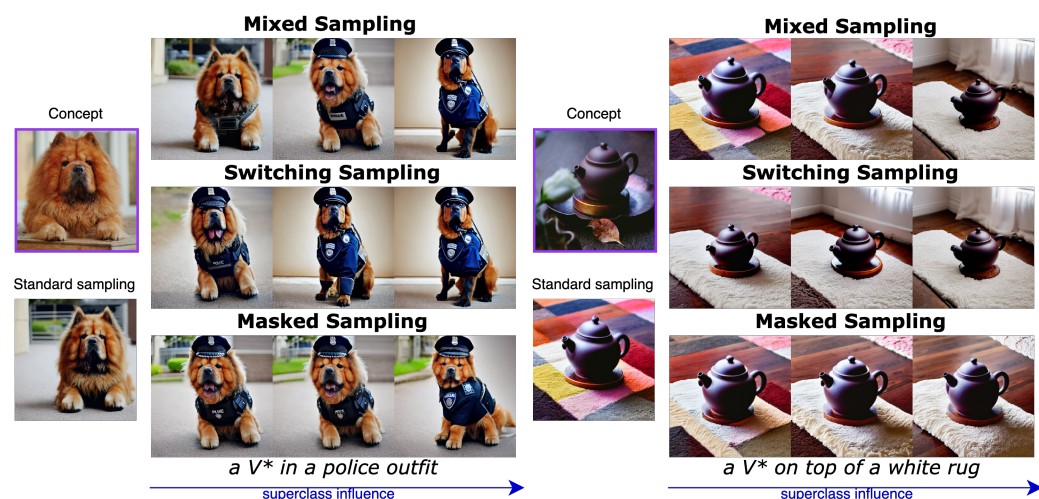

Figure 2: **Effects of Superclass Influence on Different Sampling Methods.** For Mixed Sampling, the influence is adjusted by varying the superclass guidance scale $\omega_s = [1.0, 3.5, 5.0]$ with $\omega_c = 7.0 - \omega_s$. For Switching Sampling, we vary the switching step $t_{sw} = [3, 7, 20]$ . For Masked Sampling, the mask is modified by altering the thresholding quantile $q = [0.3, 0.5, 0.9]$.

The first step is to select a fundamental fine-tune model on the basis of which we can compare different sampling techniques. For each model, we propose constructing a complete Pareto front of the Mixed sampling. We chose Mixed sampling as our baseline because it is the simplest efficient method, characterized by a single hyperparameter.

It is essential to select a model whose Pareto frontier exhibits a sufficiently large length; this allows for a clearer distinction between the varying parameters. Additionally, this front should lie within the optimal balance between concept fidelity and editability comparing to other fine-tuning methods. By doing so, we can examine sampling not only in scenarios where the model performs poorly but also ensure that sampling does not undermine performance in cases where the model excels.

Once the base model is chosen, we fix it and proceed to compare different sampling techniques. For each method, we demonstrate its behaviour at different hyperparameter values. We illustrate the optimal points with generation examples and prove our findings with user study.

## 4 EXPERIMENTS

**Dataset** For evaluation, we use the Dreambooth (Ruiz et al., 2023) dataset. It contains 30 concepts of different categories, including pets, interior decoration, toys, backpacks, etc. For each concept, we used 25 contextual text prompts, which include accessorisation, appearance and background modification. For each concept we generate 10 images per prompt. In total, there are 750 unique concept-prompt pairs and a total of 7500 images for robust evaluation.

**Evaluation Metrics** To estimate the concept identity preservation we use the Image Similarity (IS) between real and generated images as in (Ruiz et al., 2023). Higher values of this metric usually indicate better subject fidelity. However,

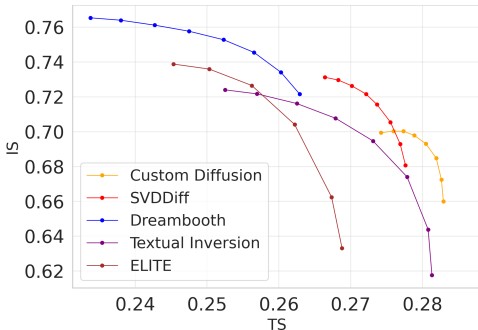

Figure 3: Mixed sampling Pareto frontiers for different fine-tuning methods.

it should be noted that the more the generated images are aligned with the contextual prompt, the less they resemble the original images. So, even if the identity of the concept is perfectly preserved the metric will be lower. To estimate the alignment between generated images and contextual prompts (TS) we calculate the CLIP similarity of the prompt and generated images (Ruiz et al., 2023; Gal et al., 2022).

**Selecting base fine-tuning model** In the initial phase, it is essential to choose a foundational fine-tuning model to facilitate the comparison of different sampling methods. To this end, we train five distinct models for each concept, implementing diverse fine-tuning parameterizations (Ruiz et al., 2023; Han et al., 2023), optimizing text embeddings (Gal et al., 2022; Kumari et al., 2023), and leveraging a pre-trained hypernetwork (Wei et al., 2023). A comprehensive description of the model training and inference procedures can be found in Appendix B.

For each model, we conduct a full evaluation of the Mixed sampling by varying the parameter $\omega_s$ within the range of 0 to 7.0, while deriving $\omega_c$ as $7.0 - \omega_s$. Figure 3 illustrates the Mixed sampling Pareto frontiers for all aforementioned methods. The method shows the expected behaviour, as the superclass guidance scale increases the text similarity improves as well, but the more diverse generation we get the more we lose on the image similarity. Notably, the results indicate that the Mixed sampling method significantly enhances text similarity across all models. Furthermore, for each model, it is feasible to select a value for $\omega_s$ such that image similarity remains relatively unchanged, while text similarity is markedly improved.

The Pareto frontier obtained from the SVDiff model achieves a favorable balance between text and image similarity; therefore, this model has been chosen for subsequent evaluations of various sampling methods.

**Computational efficiency of sampling methods** Switching sampling maintains the same number of U-Net calls and batch size as typical inference with the concept. In contrast, Mixed, Multi-stage, Masked, and Photoswap sampling require a batch size that is twice as large. Lastly, ProFusion necessitates the same batch size as Mixed sampling but performs twice as many U-Net inferences compared to all other sampling methods.

**Proposed sampling techniques analysis** In Figure 4, the Pareto frontiers for Mixed and Switching sampling are illustrated. For the Switching sampling, the curve is obtained by varying the switching step $t_{sw} = [1, 3, 5, 7, 10, 20, 30, 40]$. We observe that the Switching sampling curves lie below the Mixed sampling curve and exhibit lower values of image similarity. This indicates that Switching sampling impacts concept identity more negatively.

Additionally, we evaluated Multi-stage sampling with various hyperparameters. In Figure 4, each Multi-stage sampling curve is generated by fixing the switching step while varying the superclass guidance scale $\omega_s = [1.0, 3.0, 5.0]$. The plots reveal that the curves for Multi-stage sampling fall between the Mixed and Switching Pareto Frontiers. Only the curves with high values of the switching step cross the Pareto frontier of Mixed sampling; however, these points correspond to very low values of image similarity, thereby compromising concept identity.

Figure 5 presents the Pareto frontier for different Masked sampling hyperparameters. For all curves, we fixed the following hyperparameters: $t_{sw} = 3, \omega_s = 3.5, \omega_c = 3.5$, as these parameters correspond to the optimal point for Multi-stage and Mixed samplings. Each curve for Masked sampling is derived by varying the quantile $q = [0.3, 0.5, 0.7, 0.9]$, which controls the mask binarization threshold. Some points on these curves cross the Mixed sampling Pareto frontier, indicating an optimal balance between image and text similarity. However, this result is unstable, as the curves exhibit chaotic behavior, suggesting that the generation results are difficult to predict and that the methods require computationally intensive hyperparameter tuning. This instability can be attributed to the noisiness of the cross-attention masks, particularly in the early stages of generation (see Appendix E).

**Comparison with existing sampling methods** To fairly compare our results with Photoswap (Gu et al., 2024) and ProFusion (Zhou et al., 2023), which were initially proposed alongside fixed fine-tuning methods, we reimplemented both approaches using the same fixed SVDiff models to eliminate any influence from differing training methods.

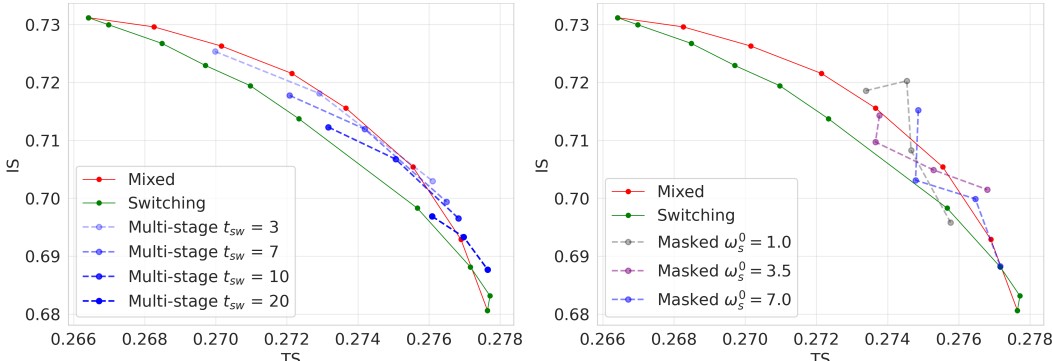

Figure 4: Pareto Frontier curves for Mixed, Switching and Multi-stage Sampling methods. Each Multi-stage sampling curve is generated by fixing the switching step while varying the superclass guidance scale $\omega_s = [1.0, 3.0, 5.0]$.

Figure 5: Pareto frontiers curves for Masked sampling. Each Masked sampling curve is derived by varying the quantile $q = [0.3, 0.5, 0.7, 0.9]$, which controls the mask binarization threshold; $t_{sw} = 3, \omega_s = 3.5$ are fixed.

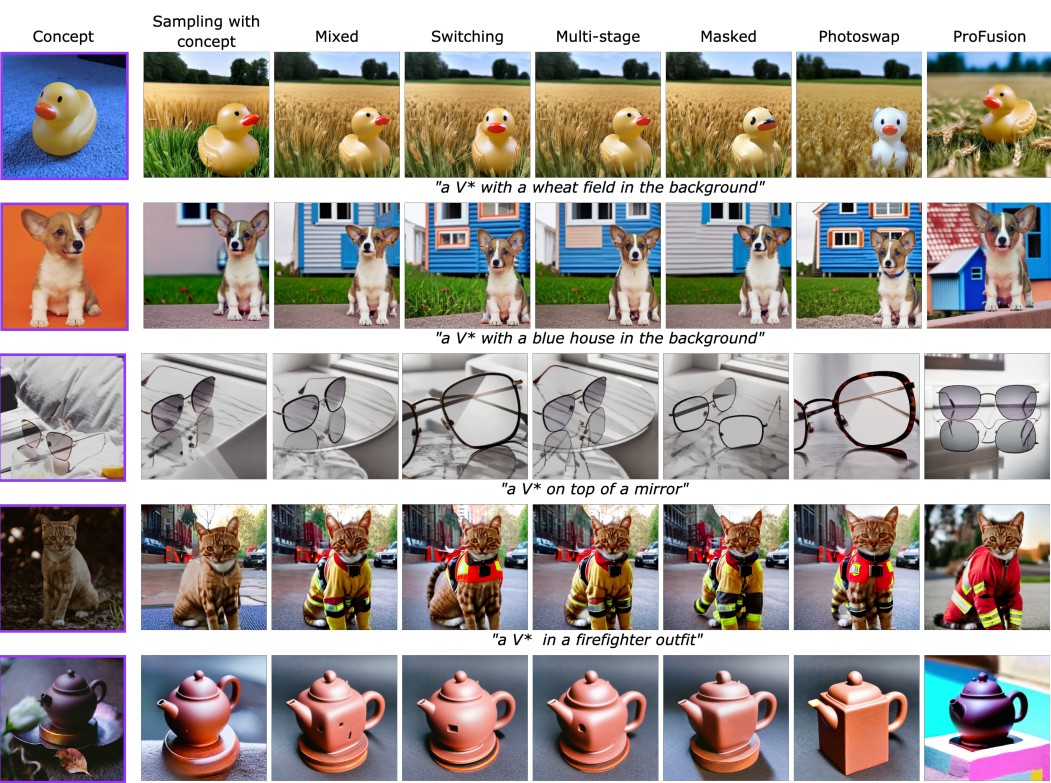

Figure 6: Examples of the generation outputs for different sampling methods.

We will first discuss the Photoswap method. In Figure 7, the Pareto front for this method is illustrated. This curve was obtained by varying three hyperparameters: $(t_{SF}, t_{CM}, t_{SM}) = [(1, 10, 15), (5, 15, 20), (10, 20, 25)]$, with the last combination representing the optimal values proposed in the original work (Gu et al., 2024). As shown in Figure 7, the curve for this method is nearly indistinguishable from that of Switching sampling. This leads us to conclude that altering the self and cross-attention maps across all layers of the U-Net affects generation almost equally as using the entire noise prediction from the superclass trajectory.

Additionally, the ProFusion Pareto frontiers are illustrated in Figure 7. Since Mixed sampling is part of the ProFusion method, we evaluated it in the same manner by fixing

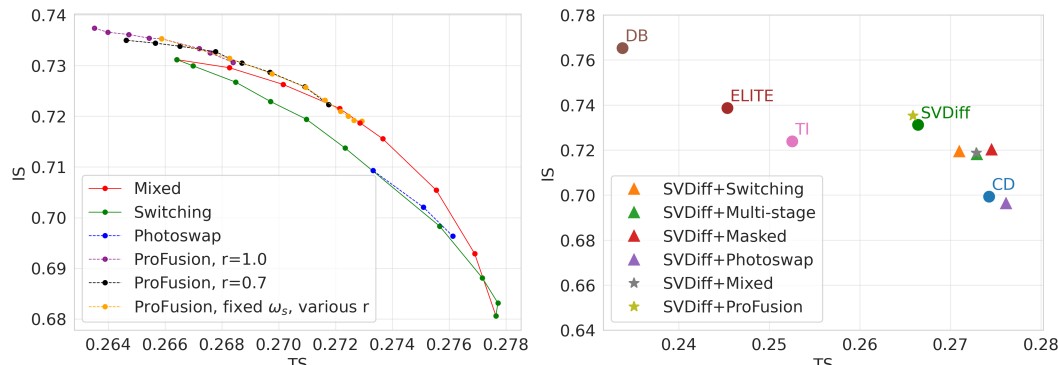

Figure 7: Pareto frontiers curves for Photoswap (Gu et al., 2024) and ProFusion (Zhou et al., 2023).

Figure 8: The overall results of different sampling methods against main personalized generation baselines.

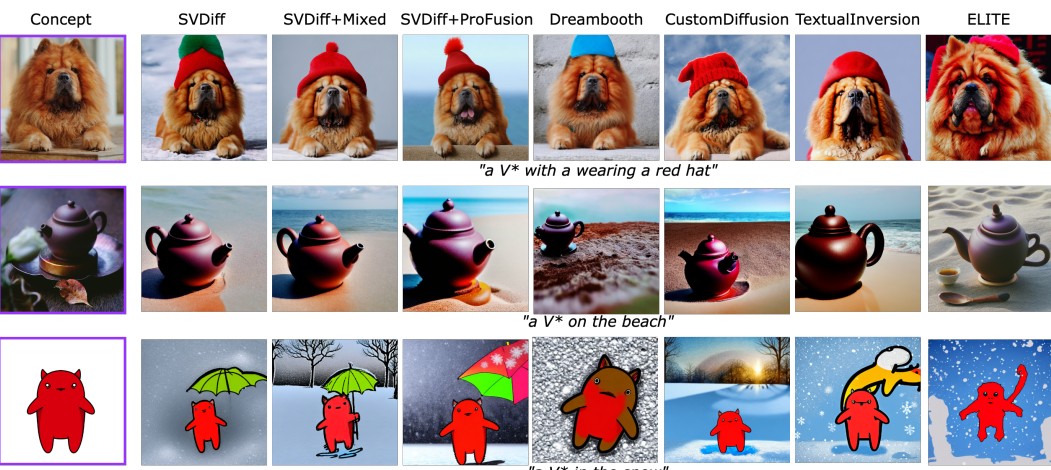

Figure 9: Examples of the generation outputs for Mixed and ProFusion sampling methods in comparison to the main personalized generation baselines.

all parameters and varying $\omega_s$. We assessed this method using two levels of fusion step intensity $r$ and constructed a distinct curve with a fixed $\omega_s = 3.5$ and various $r = [0.05, 0.1, 0.15, 0.2, 0.3, 0.4, 0.5, 0.7, 1.0]$. As observed, with decreasing fusion step intensity $r$, the curve converges more closely to the Mixed sampling curve. However, when the fusion step intensity is high, this method significantly enhances concept preservation and results in image similarity even higher than the standard sampling with concepts.

**User study** In addition to the CLIP metrics, we also conducted a human evaluation. For each sampling method we took the optimal point in terms of CLIP metric and visual generation assessment and generated 16000 pairs comparing different sampling techniques and base personalization methods (Dreambooth(DB), Custom Diffusion(CD), Textual Inversion(TI) and ELITE) with Mixed Sampling as a strong and effective baseline. See Appendix D for more details.

Given an original image of the concept, a text prompt, and 2 generated images (Mixed versus the competitor's), we asked users to answer the following questions: 1) "Which image is more consistent with the text prompt?" to evaluate text similarity 2) "Which image better represents the original image?" for image similarity 3) "Which image is generally better in terms of alignment with the prompt and concept identity preservation?" to evaluate the general impression. We provide an example of a comparison in the Appendix D.

Combining the results of the user study (Table 1) and the insights from Figure 8, which illustrates the improvements of the examined techniques against the main personalized generation baselines, we find that all sampling methods enhance the performance of the fine-tuned model in either concept or context preservation.

Table 1: User study results of the pairwise comparison of SVDDiff with Mixed sampling method versus other baselines. The values in the table show the win rate. "TS" stands for text similarity, "IS" - image similarity, and "All" corresponds to general impression.

| | SVDiff | | | | | | | | | |
|---|---|---|---|---|---|---|---|---|---|---|
| | Base | Switch | Multi-stage | Masked | Photoswap | ProFusion | DB | TI | ELITE | CD |
| TS | 0.52 | 0.51 | 0.51 | 0.51 | 0.53 | 0.49 | 0.74 | 0.67 | 0.64 | 0.51 |
| IS | 0.37 | 0.47 | 0.50 | 0.59 | 0.70 | 0.35 | 0.40 | 0.74 | 0.73 | 0.53 |
| All | 0.41 | 0.48 | 0.50 | 0.59 | 0.69 | 0.37 | 0.59 | 0.77 | 0.75 | 0.53 |

**A framework for selecting sampling method** In this section we provide an overall analysis of the performance of different sampling methods in terms of concept fidelity, alignment with text prompt and computational efficiency. In our conclusions, we rely mainly on the results of the user study, as current studies show that the CLIP metrics do not always match human perception. In case the user study doesn't reveal the difference between the performance of different methods, we draw conclusions based on the metrics and visual examples.

According to the Figure 6 standard sampling sometimes fails to align well with the text prompt. Fortunately, there are alternative sampling methods that can enhance text similarity.

As the user study and CLIP-metrics show Mixed, Switching, Multi-stage and Masked sampling show the comparable performance in terms of text similarity. The simplest and most cost-effective option is Switching Sampling. This method increases text similarity without adding to the computational load. However, sometimes it can compromise the preservation of concepts.

Mixed Sampling addresses this issue more effectively and generally provides stable results while maintaining both concept and context (see Figure 6). The trade-off is that it requires double the batch size compared to Switching Sampling.

Another viable option is Masked Sampling, which can yield better concept fidelity outcomes in situations where Mixing and Switching struggle to balance context and concept. However, it demands careful tuning of hyperparameters and may produce inconsistent results because of the cross-attention masks noisiness.

Finally, ProFusion not only enhances text similarity but also preserves a high level of concept preservation (see Figure 9), as indicated by user feedback. The downside is that it requires twice the U-Net inference compared to Mixed Sampling and require careful selection of many hyperparameters.

## 5 CONCLUSION

In this work, we investigate the role of sampling methods in enhancing personalized text-to-image generation, focusing on their interaction with fine-tuning strategies and their impact on concept fidelity and adaptability. Through systematic evaluations, we demonstrate that integrating superclass trajectories into the sampling process can lead to significant improvements, offering a flexible approach to balancing concept preservation and the ability to follow diverse textual prompts. Our analysis provides a comprehensive framework for understanding the trade-offs between different sampling techniques and their application in a variety of generative scenarios. We hope that this study will inspire further research into decoupling fine-tuning from sampling to better explore the potential of these methods independently.

Regarding the limitations of sampling techniques, we highlight two main issues. First, the sampling methods require careful tuning of hyperparameters, and finding the optimal configuration for each technique can be challenging. Second, some of the more advanced sampling techniques, such as ProFusion, come with a higher computational cost, making them less practical for real-time or large-scale applications compared to simpler alternatives.

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

## A    RELATED WORK

**Personalized Generation** Due to the considerable success of large text-to-image models Ramesh et al. (2022; 2021); Saharia et al. (2022); Rombach et al. (2022a), the field of personalized generation has been actively developed. The challenge is to customize a text-to-image model to generate specific concepts that are specified using several input images. Many different approaches Ruiz et al. (2023); Gal et al. (2022); Kumari et al. (2023); Han et al. (2023); Qiu et al. (2024); Zhou et al. (2023); Wei et al. (2023); Tewel et al. (2023) have been proposed to solve this problem and they can be divided into the following groups: pseudo-token optimization Gal et al. (2022); Zhou et al. (2023); Chen et al. (2023); Tewel et al. (2023), diffusion fune-tuning Ruiz et al. (2023); Kumari et al. (2023); Zhou et al. (2023), and encoder-based Wei et al. (2023). The pseudo-token paradigm adjusts the text encoder to convert the concept token into the proper embedding for the diffusion model. Such embedding can be optimized directly Gal et al. (2022); Tewel et al. (2023) or can be generated by other neural networks Chen et al. (2023); Zhou et al. (2023). Such approaches usually require a small number of parameters to optimize but lose the visual features of the target concept. Diffusion fine-tuning based methods optimize almost all Ruiz et al. (2023) or parts Kumari et al. (2023) of the model to reconstruct the training images of the concept. This allows to learn the input concept with high accuracy, but the model due to overfitting may lose the ability to edit it when generated with different text prompts. To reduce overfitting and the memory used, different lightweight parameterizations Han et al. (2023); Tewel et al. (2023); Hu et al. (2021) have been proposed that preserve edibility but at the cost of degrading concept fidelity. Encoder-based methods Wei et al. (2023) allow one forward pass of an encoder that has been trained on a large dataset of many different objects to embed the input concept. This dramatically speeds up the process of learning a new concept and such a model is highly editable, but the quality of recovering concept details may be low. Generally, the main problem with existing personalized generation approaches is that they struggle to simultaneously recover a concept with high quality and generate it in a variety of scenes.

**Sampling strategies** Much work has been devoted to the study of sampling for text-to-image diffusion models, not only in the task of personalized generation, but also in image editing. In this paper, we investigate a narrower question: how we can optimally combine the two trajectories on superclass and concept to simultaneously have high concept fidelity and high editability. The ProFusion paper Zhou et al. (2023) considered one way of combining these trajectories (mixed sampling), which we analyze in detail in our paper (see Section 3.1), and show its properties and problems. In ProFusion, authors additionally proposed a more complex sampling procedure, which we observed to be redundant compared to mixed

sampling, as can be seen in our experiments (see Section 4). In Photoswap Gu et al. (2024) authors consider another way of combining trajectories by superclass and concept, which turns out to be almost identical to the switching sampling strategy that we discuss in detail in Section 3.2. We show why this strategy fails to achieve simultaneous improvements in concept reconstruction and editability. In the paper, we propose a more efficient way of combining these two trajectories that achieves an optimal balance between the two key features of personalized generation: concept reconstruction and its editability.

## B  Training details

The Stable Diffusion-2-base model is used for all experiments. For the Dreambooth, Custom Diffusion and Textual Inversion methods we used the implementation from `https://github.com/huggingface/diffusers`.

**SVDiff** We implement the method based on the `https://github.com/mkshing/svdiff-pytorch`. The parametrization is applied to all text encoder and U-Net layers. The models for all concepts were trained for 1600 using Adam optimizer with batch size = 1, learning rate = 0.001, learning rate 1d = 0.000001, betas = (0.9, 0.999), epsilon = 1e-8 and weight decay = 0.01.

**Dreambooth** All query, key, value layers in text encoder and U-Net were trained during fine-tuning. The models for all concepts were trained for 400 steps using Adam optimizer with batch size = 1, learning rate = 0.001, betas = (0.9, 0.999), epsilon = 1e-8 and weight decay = 0.01.

**Custom Diffusion** The models for all concepts were trained for 1600 steps using Adam optimizer with batch size = 1, learning rate = 0.00001, betas = (0.9, 0.999), epsilon = 1e-8 and weight decay = 0.01.

**Textual Inversion** The models for all concepts were trained for 10000 steps using Adam optimizer with batch size = 1, learning rate = 0.005, betas = (0.9, 0.999), epsilon = 1e-8 and weight decay = 0.01.

**ELITE** We used pre-trained model from the official repo `https://github.com/csyxwei/ELITE` with $\lambda = 0.6$ and inference hyperparams from the original paper.

## C  Data preparation

For each concept, we used inpainting augmentations to create the training dataset. We took an original image and segmented it using the Segment Anything model on top of the CLIP cross-attention maps. Then we crop the concept from the original image, apply affine transformations to it, and inpaint the background. We used 10 augmentation prompts, different from the evaluation prompts, and sampled 3 images per prompt, resulting in a total of 30 training images per concept. We commit to open-source the augmented datasets for each concept after publication.

## D  User Study

We provide an example of a task in the user study in Figure 10. In total, we collected 48864 answers from a 200 unique users for a 16000 unique pairs. For each task, a user was presented with three questions: 1) "Which image is more consistent with the text prompt?" 2) "Which image better represents the original image?" 3) "Which image is generally better in terms of alignment with the prompt and concept identity preservation?". For each question, a user gives one of the three answers: "1", "2", or "Can't decide".

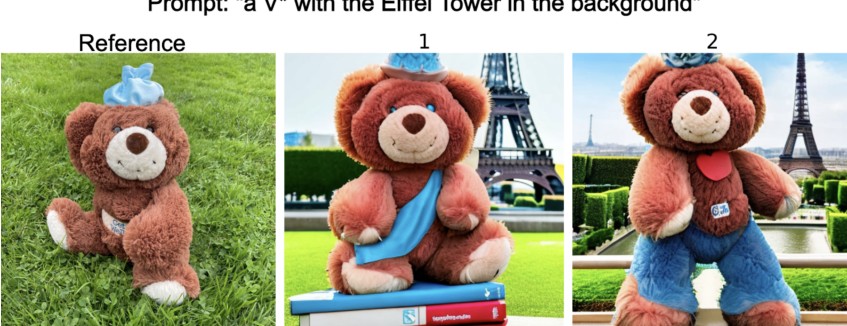

Figure 10: An example of a task in the user study

# E    CROSS-ATTENTION MASKS

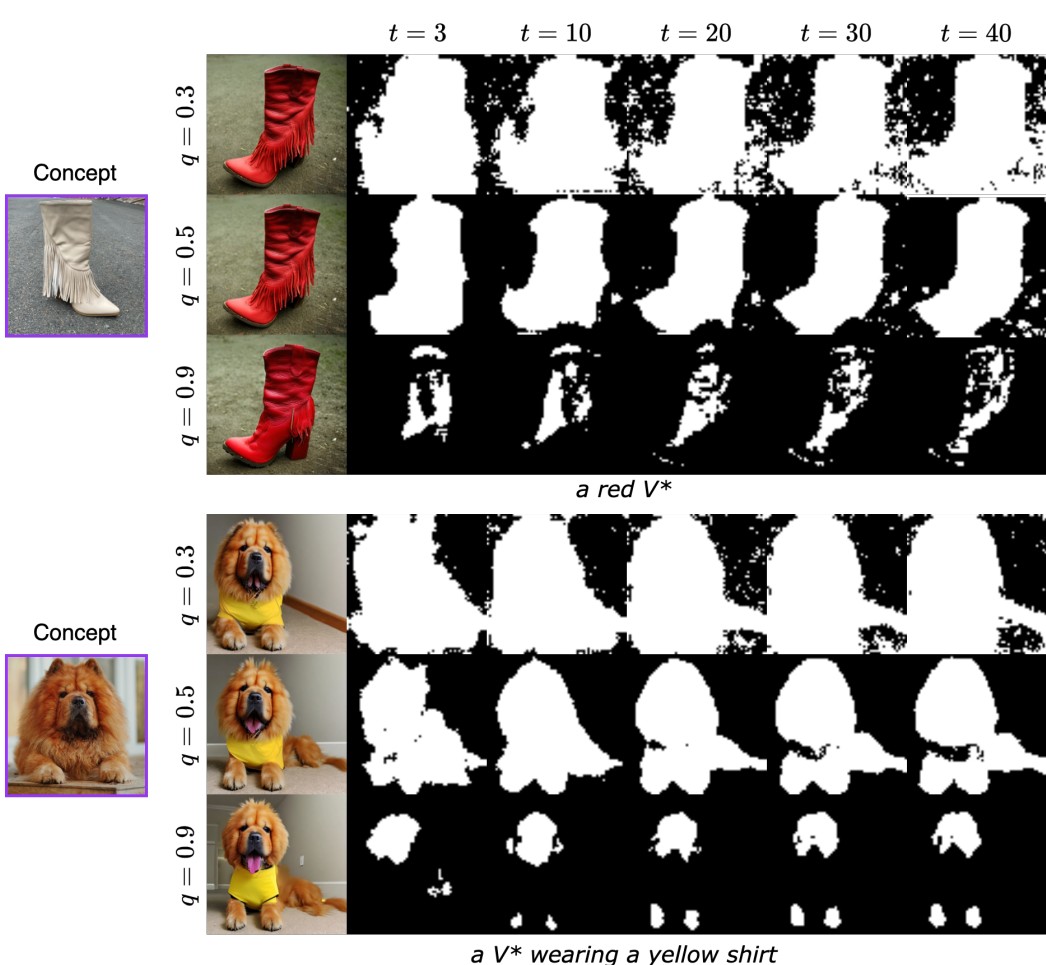

Figure 11: Visualization of the cross-attention masks for Masked sampling examples. Here, $q$ defines the thresholding quantile and $t$ the denoising step.

# F    ADDITIONAL EXAMPLES

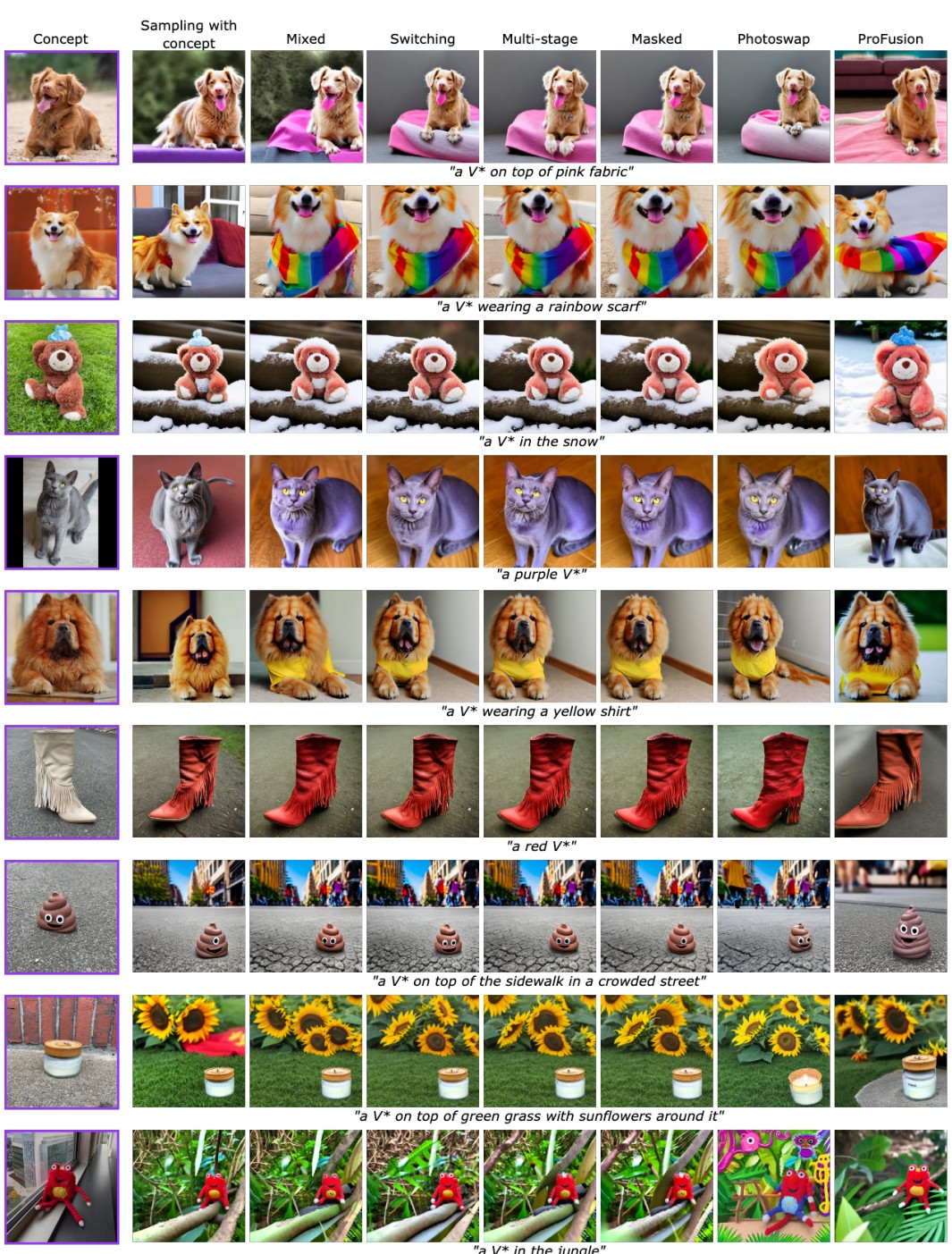

Figure 12: Additional examples of the generation outputs for different sampling methods.

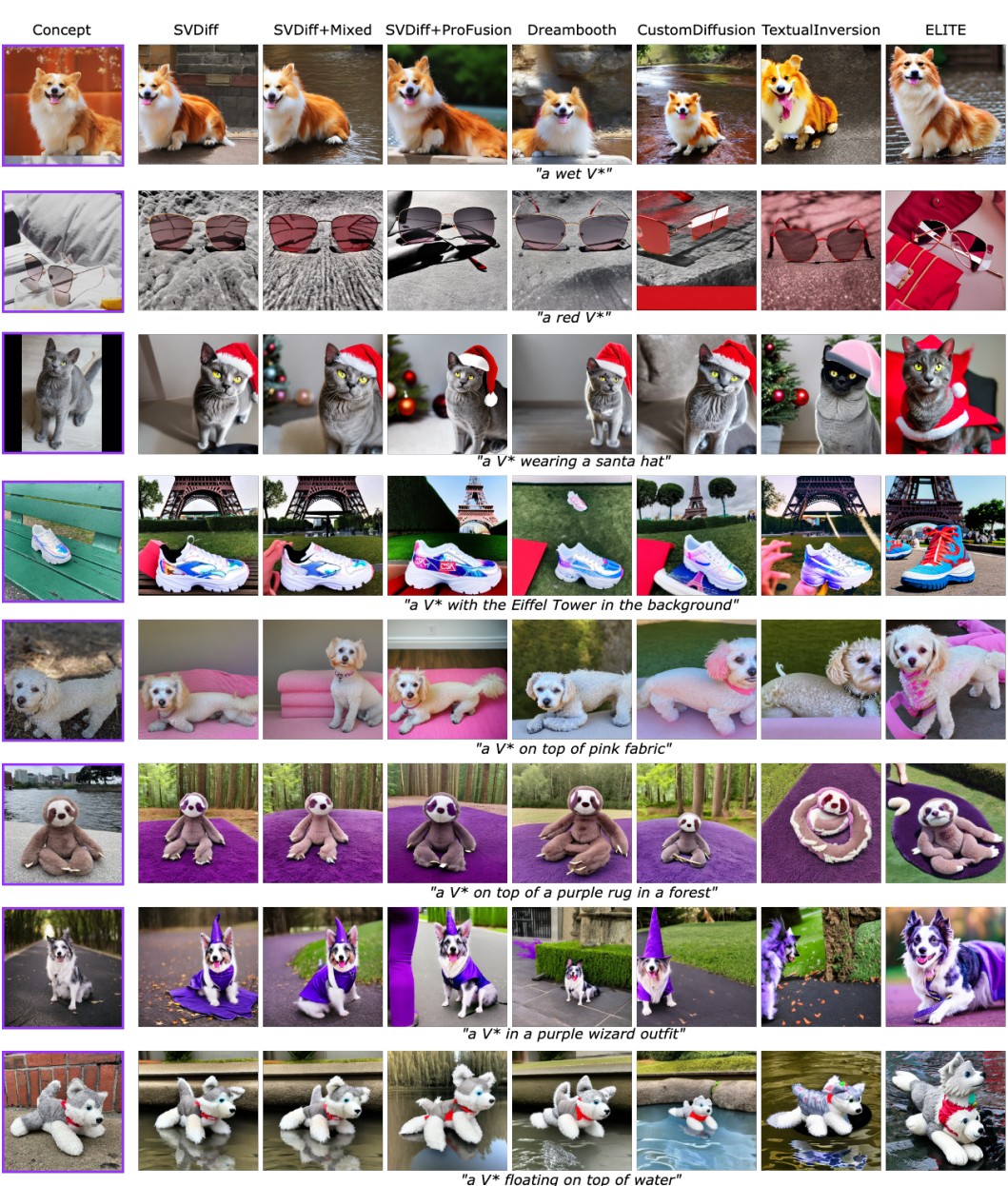

Figure 13: Additional examples of the generation outputs for Mixed and ProFusion sampling methods in comparison to the main personalized generation baselines.

## G  Dreambooth results

We conduct additional analysis of different sampling methods in combination with Dreambooth. Figure 14 shows that Mixed Sampling still overperforms Switching and Photoswap, while Multi-stage and Masked struggle to provide an additional improvement over the simple baseline. Figure 15 shows that all methods allow for improvement TS with a negligent decrease in IS while Mixed Sampling provides the best IS among all samplings.

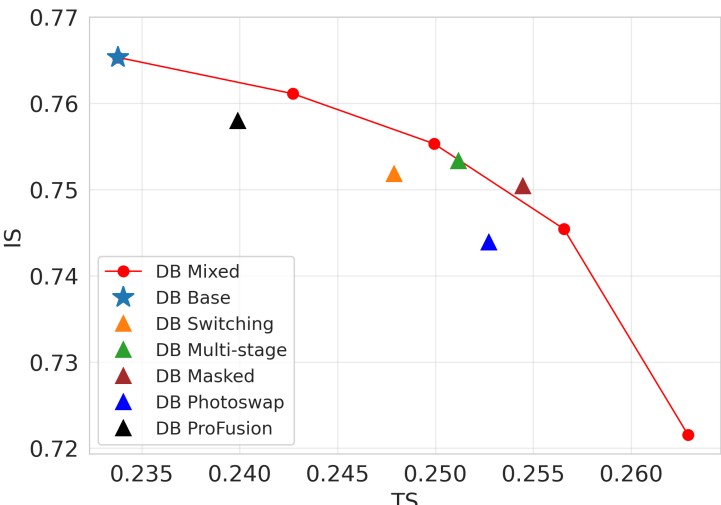

Figure 14: CLIP metrics for different sampling strategies on top of a Dreambooth fine-tuning method.

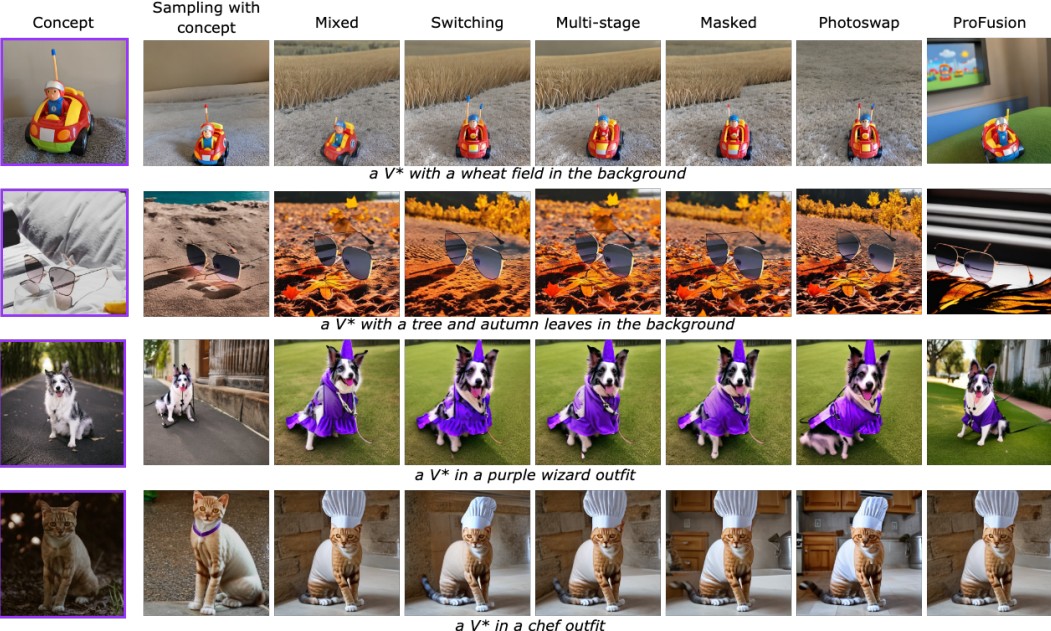

Figure 15: Additional examples of the generation outputs for different sampling methods on top of a Dreambooth fine-tuning method.

# H  Complex Prompts Setting

We conduct a comparison of different sampling methods using a set of complex prompts. For this analysis, we collected 10 prompts, each featuring multiple scene changes simultaneously, including stylization, background, and outfit:

```
live_long = [
    "V* in a chief outfit in a nostalgic kitchen filled with vintage furniture and scattered biscuit",
    "V* sitting on a windowsill in Tokyo at dusk, illuminated by neon city lights, using neon color palette",
    "a vintage-style illustration of a V* sitting on a cobblestone street in Paris during a rainy evening, showcasing muted tones and soft grays",
    "an anime drawing of a V* dressed in a superhero cape, soaring through the skies above a bustling city during a sunset",
    "a cartoonish illustration of a V* dressed as a ballerina performing on a stage in the spotlight",
    "oil painting of a V* in Seattle during a snowy full moon night",
    "a digital painting of a V* in a wizard's robe in a magical forest at midnight, accented with purples and sparkling silver tones",
    "a drawing of a V* wearing a space helmet, floating among stars in a cosmic landscape during a starry night",
    "a V* in a detective outfit in a foggy London street during a rainy evening, using muted grays and blues",
    "a V* wearing a pirate hat exploring a sandy beach at the sunset with a boat floating in the background",
]

object_long = [
    "a digital illustration of a V* on a windowsill in Tokyo at dusk, illuminated by neon city lights, using neon color palette",
    "a sketch of a V* on a sofa in a cozy living room, rendered in warm tones",
    "a watercolor painting of a V* on a wooden table in a sunny backyard, surrounded by flowers and butterflies",
    "a V* floating in a bathtub filled with bubbles and illuminated by the warm glow of evening sunlight filtering through a nearby window",
    "a charcoal sketch of a giant V* surrounded by floating clouds during a starry night, where the moonlight creates an ethereal glow",
    "oil painting of a V* in Seattle during a snowy full moon night",
    "a drawing of a V* floating among stars in a cosmic landscape during a starry night with a spacecraft in the background",
    "a V* on a sandy beach next to the sand castle at the sunset with a floaing boat in the background",
    "an anime drawing V* on top of a white rug in the forest with a small wooden house in the background",
    "a vintage-style illustration of a V* on a cobblestone street in Paris during a rainy evening, showcasing muted tones and soft grays",
]
```

The results of this comparison are presented in Figures 16, 17. We observe that basic sampling may struggle to preserve all the features specified by the prompts, whereas advanced sampling techniques effectively restore them. The overall arrangement of methods in the metric space closely mirrors that observed in the setting with simple prompts.

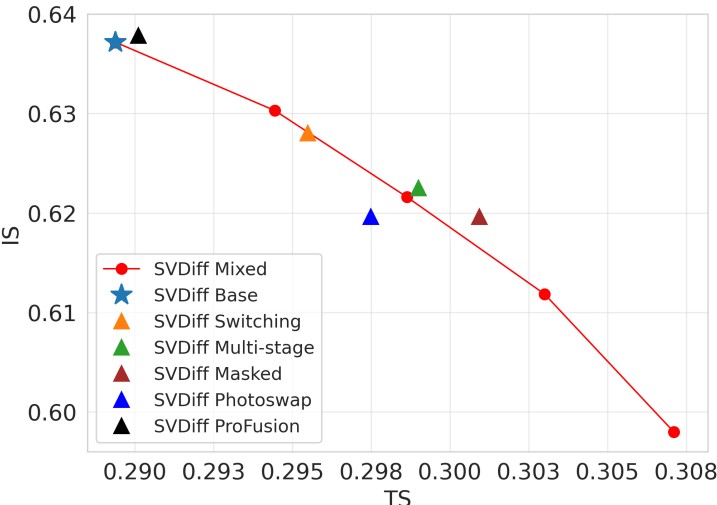

Figure 16: CLIP metrics for different sampling methods estimated on **complex prompts**.

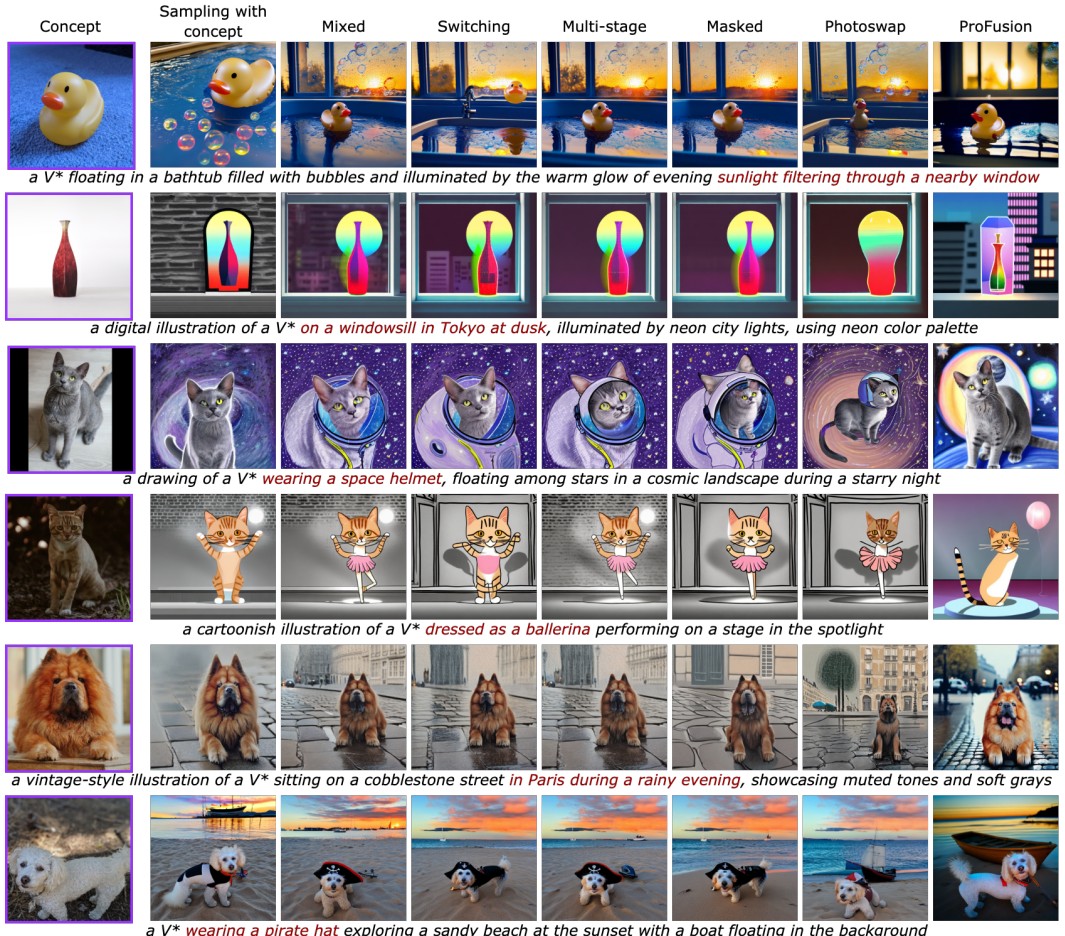

Figure 17: Additional examples of the generation outputs for different sampling methods with **complex prompts**. We highlight parts of the prompt that are missing in Sampling with concept while appearing in other methods.

## I  DINO IMAGE SIMILARITY

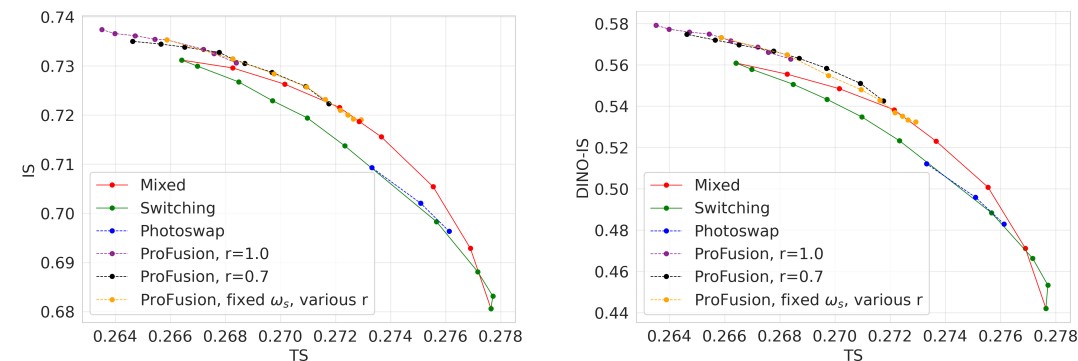

(a) Pareto frontiers curves for Photoswap (Gu et al., 2024) and ProFusion (Zhou et al., 2023).

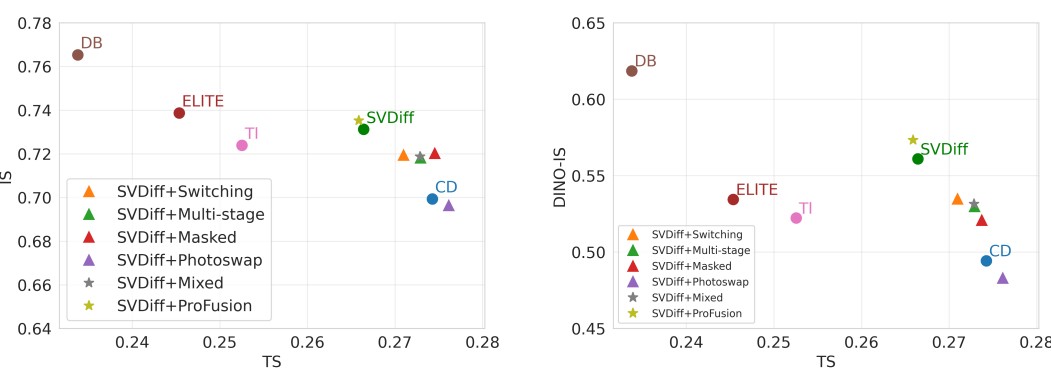

(b) The overall results of different sampling methods against main personalized generation baselines.

Figure 18: Comparison between CLIP-IS (left column) and DINO-IS (right column). We observe that despite the choice of metric, different sampling techniques and finetuning strategies have the same arrangement. The most noticeable difference is that SVDDiff superiority over EILTE and TI is more pronounced. That strengthens our motivation to select SVDDiff as the main backbone.

## J PixArt-alpha & SD-XL

We conducted a series of experiments with different backbones. For SD-XL we use SVDDiff as the finetuning method, while PixArt-alpha utilizes standard Dreambooth training. We selected hyperparameters for the Switching, Masked, and Profusion the same way we did for the experiments with SD2.

Figures 19, 20 show that Mixied Sampling follows the same pattern as for the SD2 and allows to improve TS without dramatic loss in IS. Noticeably, Mixed Sampling for SD-XL allows for improved IS and TS simultaneously. Profusion mirrors its behavior for the SD2 where it can improve IS better than Mixed Sampling while being worse at improving TS and requiring twice as many computations.

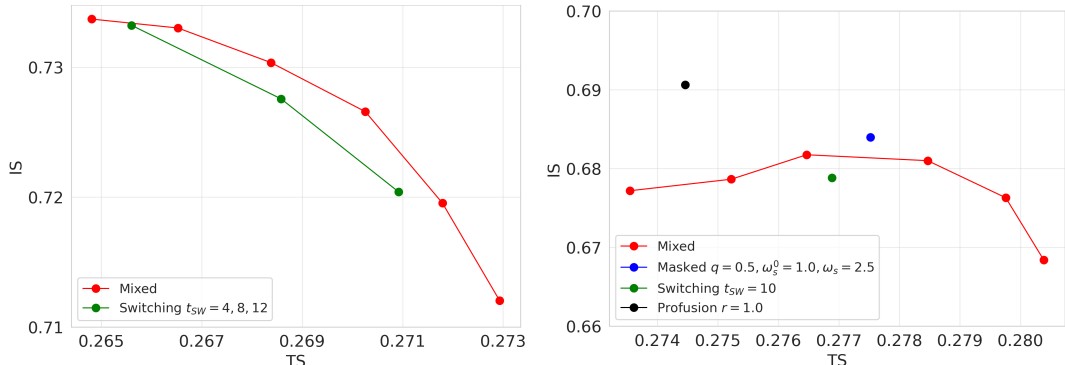

Figure 19: CLIP metrics for different sampling methods estimated on PixArt model.

Figure 20: CLIP metrics for different sampling methods estimated on SD-XL model.

## K UPDATED FIGURE 1

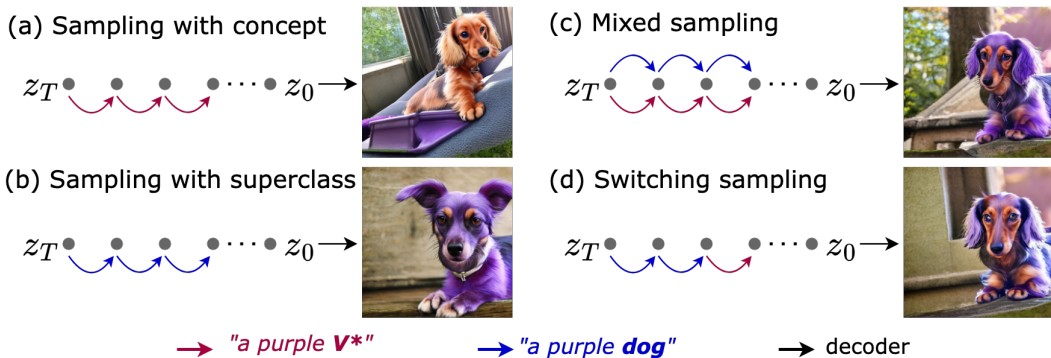

Figure 21: Updated Figure 1 for Rebuttal.