# OpenReview forum: "Beyond Fine-Tuning: A Systematic Study of Sampling Techniques in Personalized Image Generation"
_ICLR.cc/2025/Conference — Submitted to ICLR 2025_

### Official Review · Reviewer_WtXV · 2024-10-27

**Soundness:** 2
**Presentation:** 2
**Contribution:** 2
**Rating:** 5
**Confidence:** 4

**Summary:**

This paper presents a new perspective on sampling strategies in personalized image generation. The authors conduct a systematic and comprehensive analysis of the pure sampling process, which is orthogonal to pseudo-token optimization, U-Net fine-tuning, and encoder training stages typically used for learning specific concepts. They explore various methods for combining concept and superclass trajectories, including switching, mixing, and masked sampling techniques, as well as their hybrid variants. Experimental results demonstrate that integrating these strategies into existing personalized generation methods can effectively enhance both concept fidelity and context alignment.

**Strengths:**

- The paper offers a comprehensive overview and detailed analysis of various personalized generation methods, including discussions of related approaches such as ProFusion and Photoswap.
- The authors provide insights into customized generation. They propose several simple and concise sampling baselines based on the backward sampling process, which are cost-effective, require no additional training, and can be seamlessly integrated into off-the-shelf methods, independent of fine-tuning approaches.

**Weaknesses:**

- While mixed sampling, stage sampling, multi-stage sampling, and mask sampling are generally straightforward to understand, Equation (11) appears to be merely a combination of the previous methods, making it somewhat complex and less engaging. The introduction of excessive hyperparameters may reduce its practical applicability. Incorporating a complementary adaptive hyperparameter selection strategy could significantly enhance its effectiveness.
- Some of the experimental settings require more detailed explanations. Specifically, please clarify what 'TS' and 'IS' represent, and elaborate on the necessity of using the Pareto Frontier in your analysis.
- The purple dog in Fig. 1(b) is too similar to the concept dog. It is recommended to replace it with a dog that better represents the superclass to more effectively illustrate the distinctions between concept and superclass trajectories.
- Typos: Page 5 line 259 "speciall" should be "special",  Page 13, line 674: "Textual Inveersion" should be "Textual Inversion"; Page 2, line 078: "...by fine-tuning strategies; ;" should be "...by fine-tuning strategies;".

**Questions:**

- Fig. 2 shows the results of "a purple V*" and "A purple dog". Are there any changes in generating samples with "a purple V* dog"?
- In Figure 9, most of the qualitative results compare SVDiff combined with the proposed sampling methods against other state-of-the-art personalized generation methods. It would be beneficial to also compare the proposed sampling methods applied to other state-of-the-art methods. For example, comparing DreamBooth+ Mixed versus DreamBooth+ProFusion.
- In the evaluation of DreamBooth, it is mentioned using Image Similarity (IS) to assess concept identity preservation. However, DreamBooth typically employs both CLIP-I and DINO metrics to calculate the similarity between real and generated images. Please clarify which specific metric is used for IS.
- It is recommended to display the masks used in the masked sampling method to intuitively demonstrate their ability to extract concept regions and to illustrate the influence of the quantile $q$ on the mask. Additionally, please explain why you chose to use a binary mask instead of a soft mask.
- On page 4, line 199, the phrase "Up to 10 steps" requires clarification. The hyperparameter corresponding to the value 10 is not indicated as $t_{sw}$ in Figure 2. Please provide more details on this parameter.

---

> ### Author Response · Authors · 2024-11-20
>
> Thank you to the reviewer for pointing out the weaknesses.
>
> Responses to specific questions are as follows:
>
> `1. While mixed sampling, stage sampling, multi-stage sampling, and mask sampling are generally straightforward to understand, Equation (11) appears to be merely a combination of the previous methods, making it somewhat complex and less engaging. The introduction of excessive hyperparameters may reduce its practical applicability. Incorporating a complementary adaptive hyperparameter selection strategy could significantly enhance its effectiveness.`
>
> We provide Equation (11) as an aggregation of the prior methods under the unified framework. We agree that generally speaking such combination has multiple independent hyperparameters.  However, our main takeaway is that all those hyperparameters are unnecessary, and simple Mixed Sampling is optimal in terms of complexity and results. There is only one independent hyperparameter in the Mixed Sampling that should be user-defined as it controls the balance between concept fidelity and editability of the model. It should be noted that one can use Profusion to improve IS without compromising TS at the cost of doubling the computational budget.
>
> `2. Some of the experimental settings require more detailed explanations. Specifically, please clarify what 'TS' and 'IS' represent and elaborate on the necessity of using the Pareto Frontier in your analysis.`
>
> We use standard metrics to assess the quality of the personalized model. Let's denote real images of the concept and images that are generated using prompt p as
>
> $$
> I^{c} = I^{c}_{i}, i \in 1...l
> $$
>
> $$
> I^{p} = I^{p}_{j}, j \in 1...n
> $$
>
> Let's denote a "clean" prompt (i.e., prompt without superclass name and placeholder token) as $\hat{p}$. Then IS is defined as:
>
> $$
> \text{IS} = \frac{1}{l}\frac{1}{n} \sum\limits_{i,j=1}^{l,n} cos( \text{CLIP-I}(I^{c}_{i}), \text{CLIP-I}( I_j^{p} ) )
> $$
>
> And TS is defined as:
> $$
> \text{TS} = \frac{1}{n} \sum\limits_{j=1}^{n} cos( \text{CLIP-I}(I^{p}_{j}), \text{CLIP-T}(\hat{p}) )
> $$
>
> Where CLIP-I and CLIP-T denote CLIP image and text embeddings.
>
> The Pareto Frontier appears naturally as a balance between concept fidelity and editability. We use that notion to show that different methods, despite their differences, usually lie below or on the curve that is defined by Mixed Sampling.
>
> `3. The purple dog in Fig. 1(b) is too similar to the concept dog. It is recommended to replace it with a dog that better represents the superclass to more effectively illustrate the distinctions between concept and superclass trajectories.`
>
> Thank you for you recommendation, we attached an updated version of the Figure 1 in Appendix K.
>
> `4. Typos: Page 5 line 259 "speciall" should be "special", Page 13, line 674: "Textual Inveersion" should be "Textual Inversion"; Page 2, line 078: "...by fine-tuning strategies; ;" should be "...by fine-tuning strategies;".`
>
> Thank you for your comment; we will correct these typos in the next version of the paper.
>
> `5. Fig. 2 shows the results of "a purple V*" and "A purple dog". Are there any changes in generating samples with "a purple V* dog"?`
>
> We need to clarify the exact way to generate images using different models. ELITE and Textual Inversion use "a purple V*" to generate samples, while DreamBooth, Custom Diffusion, and SVDDiff use "a purple V* dog". We simplify our notation by using the same prompt "a purple V*" across all models, even if the generation uses a superclass name inside the prompt. It's worth noting that this difference doesn't affect the way the Text Similarity is evaluated. We use the "empty" prompt without a superclass name or placeholder token (i.e., "a purple") to compute CLIP-T embeddings that are used to compute the metric.

---

> > ### Author Response · Authors · 2024-11-20
> >
> > `6. In Figure 9, most of the qualitative results compare SVDiff combined with the proposed sampling methods against other state-of-the-art personalized generation methods. It would be beneficial to also compare the proposed sampling methods applied to other state-of-the-art methods. For example, comparing DreamBooth+Mixed versus DreamBooth+ProFusion.`
> >
> > We fill this gap in our analysis by providing qualitative and quantitative results for the combination of different sampling methods with DreamBooth. Appendix G shows that the behavior of different samplings is similar to SVDDiff model. Also, we show that Mixed Sampling on top of ELITE, TI, and CD produces similar results as for SVDDiff (see Figure 3).
> >
> > `7. In the evaluation of DreamBooth, it is mentioned using Image Similarity (IS) to assess concept identity preservation. However, DreamBooth typically employs both CLIP-I and DINO metrics to calculate the similarity between real and generated images. Please clarify which specific metric is used for IS.`
> >
> > We selected CLIP-I as the main metric of Image Similarity in all our experiments. However, we analyzed the behavior of different methods using DINO embeddings and found no significant difference between them except for the different value ranges. We provide Figures 18(a) in Appendix I similar to Figures 7, and 8 except that DINO IS is used.
> >
> > `8. It is recommended to display the masks used in the masked sampling method to intuitively demonstrate their ability to extract concept regions and to illustrate the influence of the quantile $q$ on the mask. Additionally, please explain why you chose to use a binary mask instead of a soft mask.`
> >
> > We display several masks depending on the quantile $q$ in the Appendix E. They show the ability to extract the concept region even for the first denoising steps.
> >
> > The motivation behind binarization is that soft masks are very noisy, especially at the first steps, and changing guidance for the random pixels outside the concept region could harm the final results. Instead, we try to separate the concept from the background as exactly as we can and apply different guidance for the background and concept.
> >
> > `9. On page 4, line 199, the phrase "Up to 10 steps" requires clarification. The hyperparameter corresponding to the value 10 is not indicated as in Figure 2. Please provide more details on this parameter.`
> >
> > We meant that according to the Figure 2, a switching step of 3 or 7 (both less than 10) is sufficient to restore text alignment, which is inadequate in basic sampling.

---

> ### Author Response · Authors · 2024-11-28
>
> Dear Reviewer,
>
> Thank you for your valuable feedback and the time you have invested in reviewing our submission. If you have any additional questions, concerns, or need further clarification, we would be happy to continue the discussion.
>
> Thank you once again for your attention. We look forward to your response.

---

### Official Review · Reviewer_6r26 · 2024-11-01

**Soundness:** 2
**Presentation:** 2
**Contribution:** 2
**Rating:** 3
**Confidence:** 4

**Summary:**

The paper performs a detailed ablation study around different sampling approaches for personalized text-to-image diffusion models. The different sampling approaches are built around sampling with the concept (e.g., "photo of a V*") and sampling with the superclass of the given concept (e.g., "photo of a dog") and compared various approaches of combining those two during sampling (e.g., by combining them via classifier-free guidance or by doing some sampling steps with the concept prompt and some sampling steps with the superclass prompt). The ablations show that, given a personalized model, the sampling strategy will affect how much the resulting image will follow the overall prompt and how much it preservest the identity.

**Strengths:**

This paper performs a detailed ablation study of various sampling strategies combining the personalized prompt (e.g., photo of a V*) and the superclass prompt (e.g., photo of a dog). Many different sampling approaches are evaluated across many different hyperparameters across different personalized models and the results are presented in graphs, showing the pareto front of the different sampling approaches and hyperparameters.

The ablations show that the sampling strategy and sampling parameters clearly have an impact on the quality of the output. As such, the paper highlights that choosing a good sampling strategy and sampling hyperparameters is crucial for personalized text-to-image models and that related works should disentangle the impact of improved sampling strategies from the impact of potential other changes to the personalization approach (e.g., architecture or finetuning changes).

**Weaknesses:**

While the ablation is comprehensive the results are not necessarily surprising and it is also not clear to me what to do with the results. Different sampling approaches have different advantages and trade-offs (both in quality and sampling cost) and there is no winner. Rather, the best sampling approach is dependent on the model, desired outcome, and available inference compute. But I feel like that was clear before this and is also evidenced by the large related literation on sampling approaches for diffusion models, all of which have different trade-offs and advantages.
So it's not clear what novel knowledge this ablation concretely contributes or how it would change the approach to current personalization approaches.

**Questions:**

What is the concrete outcome of the ablations? Do you suggest to use a default sampling approach or specific hyperparameters? Will that recommendation translate to other models besides SVDDiff?

---

> ### Author Response · Authors · 2024-11-20
>
> Thank you to the reviewer for pointing out the weaknesses.
>
> Responses to specific questions are as follows:
>
> `1. While the ablation is comprehensive the results are not necessarily surprising and it is also not clear to me what to do with the results. Different sampling approaches have different advantages and trade-offs (both in quality and sampling cost) and there is no winner. Rather, the best sampling approach is dependent on the model, desired outcome, and available inference compute. But I feel like that was clear before this and is also evidenced by the large related literation on sampling approaches for diffusion models, all of which have different trade-offs and advantages. So it's not clear what novel knowledge this ablation concretely contributes or how it would change the approach to current personalization approaches.`
>
> While there is a wealth of literature on sampling methods for diffusion models, their effectiveness in personalized generation is not thoroughly examined. A few studies propose test-time sampling, frequently merging it with fine-tuning approaches, model configurations, and fixed hyperparameters (ProFusion, Photoswap). However, these studies often overlook detailed ablation analyses that explain the contributions made by sampling and its distinguished components and the selection of hyperparameters.
>
> As a result of our work, we can identify exactly what differences sampling methods have and what kind of improvement we can expect using them. Moreover, we observe that those insights can be extended to various finetuning methods like TI, DB, CD, and ELITE (see Figure 3, Appendix G) and architectures (see Appendix J).
>
> `2. What is the concrete outcome of the ablations? Do you suggest to use a default sampling approach or specific hyperparameters? Will that recommendation translate to other models besides SVDDiff?`
>
> The main outcome of the ablation is that we establish Mixed Sampling as the cheapest and most efficient way to balance concept fidelity with concept editability. We suggest using Mixed Sampling in cases where higher TS is required and Profusion when a higher IS is necessary. This result can be translated to other finetuning methods (see Appendix G) and to other architectures (see Appendix J).
>
> We aim to encourage researchers to benchmark their work against naive techniques such as Mixing, ensuring that the complexity of newly proposed sampling strategies is well justified.

---

> ### Author Response · Authors · 2024-11-28
>
> Dear Reviewer,
>
> Thank you for your valuable feedback and the time you have invested in reviewing our submission. If you have any additional questions, concerns, or need further clarification, we would be happy to continue the discussion.
>
> Thank you once again for your attention. We look forward to your response.

---

### Official Review · Reviewer_JY4n · 2024-11-02

**Soundness:** 3
**Presentation:** 4
**Contribution:** 2
**Rating:** 6
**Confidence:** 5

**Summary:**

Personalizing text-to-image models involves fine tuning a generative model on a small set of images that depict a new concept that wasn’t part of the training data of the text-to-image model. There are two objectives for every personalization method, (1) to be able to generate images that faithfully resemble the source subject and (2) to maintain the ability to generate images from novel prompts. The authors study the effectiveness of test-time sampling strategies for improving personalization methods. While there have been papers that developed or used various sampling techniques for aligning the personalized models in test-time, this paper studies different sampling techniques and compares them in terms of running time, subject fidelity and prompt alignment.

Test-time sampling strategies usually run the denoising prediction of the model on the super-class concept, or on the new personal concept (and sometimes both) and make use of this to steer the generation process to either be more prompt-aligned or subject aligned. The ultimate good is to balance the trade-off of text- and subject fidelity.

The paper considers three simple-strategies, mixed-, switching- and masked- strategies. They also consider two strategies developed in the ProFusion method and Photoswap. For evaluation, the CLIP score and image-similarity metrics were used. The paper shows mixed-sampling consistently improving base personalization methods, while switching harms subject-fidelity. Masked-Sampling on the other hand induces unstable behavior for different hyperparameters.

**Strengths:**

[1] The paper is well written and the experimental setup follows existing evaluation protocols

[2] The authors use the common metrics in their evaluation (CLIP Score and Image Similarity), and use publicly available dataset.

[3] The authors run their evaluation on multiple baseline personalization methods including DreamBooth, TI and SVDiff.

**Weaknesses:**

[1] The main concern I have about this paper is that it doesn’t convince the reader that test-time sampling strategies are effective and necessary for optimal personalization results. In particular, altering the sampling strategy at test-time indicates the personalization method used is not optimal, otherwise, the model should retain its capabilities in generating images from novel prompts. The gain using different sampling strategies is not significant, at least from the qualitative and quantitative results.

[2] While the experimental section is well carried - it lacks deeper analysis beyond CLIP and Image similarity report. What would be the result if you change the sampling with superclass to something else,  how would that change the findings of the results ? For example instead of sampling “A dog wearing a police outfit”-> “Wearing a police outfit”. Another interesting thing to check is to measure how different the “super-class” trajectories are from the personalized trajectory, and whether it indicates something regarding the personalized model.

[3] The authors should consider new state-of-the-art personalization methods and compare the effectiveness of using test-time sampling strategies on newer methods.

[4] The qualitative samples don’t show enough evidence that test-time sampling strategies are sufficient for ultimate personalization results. It is necessary to assess the performance on long prompt and complicated prompts.

**Questions:**

As appears in the paper, the authors use the fine-tuned model prediction on the super-class rather than taking the base-model prediction. Is this the case ? and if so, why not use the base-model prediction as it has better text-alignment.

---

> ### Author Response · Authors · 2024-11-20
>
> Thank you to the reviewer for pointing out the weaknesses.
>
> Responses to specific questions are as follows:
>
> `1. The main concern I have about this paper is that it doesn’t convince the reader that test-time sampling strategies are effective and necessary for optimal personalization results. In particular, altering the sampling strategy at test-time indicates the personalization method used is not optimal, otherwise, the model should retain its capabilities in generating images from novel prompts. The gain using different sampling strategies is not significant, at least from the qualitative and quantitative results.`
>
> Despite the current personalization methods achieving high results, a common issue is that the balance between concept identity and prompt alignment can become disrupted in certain cases [1, 2]. While new personalization techniques aim to establish a more optimal balance, the example of SVDDiff illustrates that even an optimal position in the TS/IS plane does not guarantee satisfactory results with base sampling (see Figure 6). In such cases, advanced sampling strategies can effectively enhance these samples without requiring model retraining. To further substantiate this claim, we provide additional examples in which base sampling fails to align with the text, while more advanced sampling techniques significantly improve the outcome (see Appendix, Figure 12).
>
> [1] CoRe: Context-Regularized Text Embedding Learning for Text-to-Image Personalization. Feize Wu, Yun Pang, Junyi Zhang, Lianyu Pang, Jian Yin, Baoquan Zhao, Qing Li, Xudong Mao.
>
> [2] PALP: Prompt Aligned Personalization of Text-to-Image Models. Moab Arar, Andrey Voynov, Amir Hertz, Omri Avrahami, Shlomi Fruchter, Yael Pritch, Daniel Cohen-Or, Ariel Shamir.
>
> `2. While the experimental section is well carried - it lacks deeper analysis beyond CLIP and Image similarity report.`
>
> We agree that more elaborate metrics for personalized generation exist; however, CLIP Image Similarity and Text Similarity are the most widespread. We complement CLIP Image Similarity with DINO Image Similarity in Appendix I (DINO Image Similarity) but find practically no differences.
>
> Moreover, we provide a User Study that supports our findings that Mixed Sampling is the strong baseline for personalized generation, outperforming some of the more complex methods like Photoswap using the same or smaller computational budget.
>
> `3. The authors should consider new state-of-the-art personalization methods and compare the effectiveness of using test-time sampling strategies on newer methods.`
>
> In our experiments, we utilized core personalization models with open-source code to demonstrate that test-time sampling methods are independent of the fine-tuning approach and can be successfully transferred across various parameterizations (Figure 3). If you have specific methods you would like us to include in our experiments, can you please specify them, we will do our best to fulfill your requests.

---

> > ### Author Response · Authors · 2024-11-20
> >
> > `4. The qualitative samples don’t show enough evidence that test-time sampling strategies are sufficient for ultimate personalization results. It is necessary to assess the performance on long prompt and complicated prompts.`
> >
> > We agree that it is important to check whether our findings hold for the challenging prompts. We provide such analysis in Appendix H (Complex Prompts Setting). To summarize, we found the same behavior of the sampling methods in this case.
> >
> > Also, we would like to point out that our work doesn't try to provide the ultimate personalization method but analyzes existing approaches in the highly controlled setup and points out their limitations and drawbacks. The ultimate personalization method would require improving IS and TS at the same time, and our findings show that none of the methods can provide results that will significantly exceed the Pareto Frontier defined by Mixed Sampling. We propose a framework to choose the method based on the demands: Mixing Sampling as a simple baseline that can improve TS at the cost of a slight decrease in IS, and Profusion as a computationally expensive way to increase IS while preserving TS.
> >
> > `5. What would be the result if you change the sampling with superclass to something else, how would that change the findings of the results ? For example instead of sampling “A dog wearing a police outfit” $\rightarrow$ “Wearing a police outfit”. Another interesting thing to check is to measure how different the “super-class” trajectories are from the personalized trajectory, and whether it indicates something regarding the personalized model. ... As appears in the paper, the authors use the fine-tuned model prediction on the super-class rather than taking the base-model prediction. Is this the case ? and if so, why not use the base-model prediction as it has better text-alignment.`
> >
> > The selection of the "superclass" trajectory to incorporate in the sampling process is a hyperparameter (i.e. from a finetuned model or from the base model, with superclass token or not) that we fix across all sampling methods to ensure a fair comparison. We believe that altering this hyperparameter simultaneously for all methods will not significantly impact their relative performance, and our main conclusions will remain valid.
> >
> > We chose to focus on the superclass trajectory from the fine-tuned model as it is the least expensive way to get a different enough but still well-aligned trajectory. Moreover, the prior work uses the same approach, so we decided to analyze it.
> >
> > Although selecting the best "superclass" trajectory is out of the scope of our work, we will provide some analysis in the future revision of the paper.

---

> ### Comment · Reviewer_JY4n · 2024-11-26
> **Thank you for the detailed response.**
>
> I will raise my score to (6) - I think the paper is well written, the experiments are thorough and comprehensive, which gives a good insight about the trade-off between test-time sampling methods. Nonetheless, I think the paper lacks deeper insights.
>
> Re 1.
> Please add the missing references in the paper - seems like they are relevant.
>
> Re 2.
> I am aware that CLIP And Image Similarity metrics are the standard in this domain. My take-a-way message from the paper is that it is useful to use Mixing strategy to improve text-alignment of the personalisations methods. This finding is not surprising, and it has been proposed before (see [1] for example). Nonetheless, I do appreciate the authors efforts in doing thorough comparison between different methods.
> The authors are encouraged to provide deeper analysis in a follow-up/revised version - some examples for questions I have regarding test-time sampling strategies:
> * Can you quantify the dis-similarity between the super-class  and personalized- trajectories ? Even for the simple case-study, how the trajectories for the prompt="A photo of [V]" is different than that for the prompt="A photo of [Super-class]". Clearly the more they trajectories are different, the more likely the model overfits the training images. Will the Mixing strategy still work in the extreme case where the two trajectories are different (e.g., if the personalized model overfits the input data).
>
> [1] https://arxiv.org/abs/2307.06925
>
>
> Re 4:
> Thanks for taking the time to do the additional experiments.
>
> Re 5:
> Thank you for justifying your choice.

---

> ### Author Response · Authors · 2024-11-28
>
> Thank you for raising the score. We appreciate that you acknowledged our thorough and comprehensive experiments and found our paper well-written.
>
> Regarding the dissimilarity of the trajectories, Figure 3 illustrates the full plots for mixing sampling, including the border points. On the far left, we show the sampling using only the concept, while on the far right, we use only the superclass for sampling. We observe that the differences between the trajectories are strongly model-dependent. For instance, in the case of Textual Inversion, where the model remains unchanged and only the embedding is trained, this difference is significant. In contrast, when we alter the model weights, the difference diminishes.
>
> Significant overfitting can reduce the pronounced effect of sampling. On the other hand, blending with the trajectory of a non fine-tuned model may significantly expand the Pareto frontier; however, it might introduce artefacts during the blending process due to the substantial differences in trajectories.
>
> We will take your comments into account and endeavour to carefully investigate these effects in future versions.

---

### Official Review · Reviewer_E8xN · 2024-11-11

**Soundness:** 3
**Presentation:** 3
**Contribution:** 2
**Rating:** 5
**Confidence:** 3

**Summary:**

This manuscript systematically studies the sampling techniques for personalized text-to-image diffusion models. Various combinations of concept and superclass trajectories are explored. A comprehensive analysis of existing techniques is presented that covers fidelity, edit ability, and computational efficiency. A framework is provided to determine which sampling method to use for different scenarios.

**Strengths:**

1. A comprehensive study covering various sampling techniques and different aspects of quality evaluation is presented.
2. A finetuning-independent evaluation is presented to demonstrate the effectiveness of different sampling strategies.
3. A guideline on how to choose sampling strategies is provided that can balance between image similarity, text similarity, and computation overhead.

**Weaknesses:**

1. The main weakness is in the contributions. Although this manuscript presents a comprehensive study of various sampling strategies, they are all based on existing techniques. There are not many new ideas presented in the manuscript.
2. The results shown in the paper are all easy cases with common objects. It is not sure if the conclusion will stay the same for more challenging cases involving unique objects.
3. All experiments are based on a single diffusion model backbone (Stable diffusion 2) and fine-tuning method SVDiff. It is not sure if the conclusion will still hold for other backbones and finetuning methods

**Questions:**

would these sampling strategies also be helpful for non-finetuning personalized text-to-image diffusion models, for example ELITE?

---

> ### Author Response · Authors · 2024-11-20
>
> Thank you to the reviewer for pointing out the weaknesses.
>
> Responses to specific questions are as follows:
>
> `1. The main weakness is in the contributions. Although this manuscript presents a comprehensive study of various sampling strategies, they are all based on existing techniques. There are not many new ideas presented in the manuscript.`
>
> Even though the techniques presented in the paper already exist, there is no literature providing their fair comparison in the general formulation. Furthermore, these techniques are often presented in complex combinations with others (such as the refining step utilized in ProFusion or the self-, cross- attention map replacement featured in Photoswap) and varying fine-tuning strategies.
>
> In response to this gap, our research offers a more fair comparison by standardizing a single fine-tuning parameterization across all methods. We also decompose each sampling strategy, breaking it down into less complicated components. Our findings reveal that basic techniques like Mixing and Switching can serve as strong baselines that are effective and often sufficient for addressing text misalignment while maintaining concept identity. This analytical approach stands as a noteworthy contribution to the existing body of knowledge.
>
> Our study aims to encourage researchers to benchmark their work against these naive techniques, ensuring that the complexity of newly proposed sampling strategies is well justified. Additionally, we advocate for fair comparisons by standardizing fine-tuning methods.
>
>
> `2. The results shown in the paper are all easy cases with common objects. It is not sure if the conclusion will stay the same for more challenging cases involving unique objects.`
>
> We selected the Dreambooth dataset for its widespread use in the field which helps to contextualize our results with the prior works. Some concepts represent unique objects like `can` or `rc\_car` that should be challenging enough to find in the generated images of the source model. Can you specify what classes you think is important to analyze?
>
> We agree that it is important to check whether our findings hold for the challenging prompts. We provide such analysis in Appendix H (Complex Prompts Setting). To summarize, we found the same behavior of the sampling methods in this case.
>
> `3. All experiments are based on a single diffusion model backbone (Stable diffusion 2) and fine-tuning method SVDiff. It is not sure if the conclusion will still hold for other backbones and finetuning methods`
>
> As for the finetuning methods, we have already analyzed some methods like Textual Inversion, Dreambooth, Custom Diffusion, and ELITE in combination with Mixed sampling in our paper (see Figure 3 in the paper). However, to fill the gap we provide an analysis of other samplings in combination with DreamBooth in Appendix G (Dreambooth Results) and show that our results hold for more simple methods than SVDDiff.
>
> Also, we provide results (Appendix J) for the Stable Diffusion XL and Pix-Art to show that even completely different architectures like Visual Transformer still follow the same patterns as the simple and easy-to-train Stable Diffusion 2.0.
>
> `4. Would these sampling strategies also be helpful for non-finetuning personalized text-to-image diffusion models, for example ELITE?`
>
> In Figure 3, we present the results of applying mixed sampling to ELITE. The overall impact on the metrics is similar for this model as it is for fine-tuning-based models.

---

> ### Author Response · Authors · 2024-11-28
>
> Dear Reviewer,
>
> Thank you for your valuable feedback and the time you have invested in reviewing our submission. If you have any additional questions, concerns, or need further clarification, we would be happy to continue the discussion.
>
> Thank you once again for your attention. We look forward to your response.

---

### Author Response · Authors · 2024-11-20

We are sincerely grateful to the reviewers for dedicating their time and effort to reviewing our work. We address each reviewer's comments in detail below. We have made numerous updates to the submission appendix, most notably with the results for other architectures like PixArt and SD-XL (Appendix J), for Dreambooth in combination with different sampling methods (Appendix G), and for the challenging prompts beyond the Dreambooth dataset (Appendix H).

---

### Author Response · Authors · 2024-11-25

Dear Reviewers,

We hope that we have sufficiently addressed the concerns raised in both our responses and the revised manuscript. In light of the additional evidence and clarifications provided, we would appreciate your thoughtful reconsideration of the current evaluation of this work. Should there be any remaining questions or points requiring further clarification, please feel free to let us know, and we will be happy to provide any additional information.

---

### Author Response · Authors · 2024-12-04

Dear Reviewers,

Thank you for your valuable feedback and for the time you dedicated to reviewing our work. Your insights have been instrumental in shaping the final version of our submission.

We regret that three out of the four reviewers did not respond to our updates. We would like to extend our gratitude to Reviewer JY4n for acknowledging our rebuttal and for appreciating how our experiments provide valuable insights into the trade-offs between test-time sampling methods.

In light of this, we would like to summarize some key insights from our paper and the main updates we introduced during the rebuttal.

Currently, there is no literature providing a fair comparison and comprehensive investigation of the test-time sampling improvement in the subject-driven generation. Furthermore, sampling techniques are often presented in complex combinations (such as the refining step utilized in ProFusion or the self-, cross- attention map replacement featured in Photoswap) and varying fine-tuning strategies. In response to this gap, our research offers a more fair comparison by standardizing a single fine-tuning parameterization across all methods. We also decompose each sampling strategy, breaking it down into less complicated components.

Our rebuttal updates substantiate the main result of the paper: Mixing Sampling is a simple baseline that can improve TS at the cost of a slight decrease in IS, and Profusion is a computationally expensive way to increase IS while preserving TS. This claim holds for a wide variety of prompts, different architectures, and fine-tuning methods.

**Appendix E: Cross-Attention Masks** as requested by Reviewer WtXV

&nbsp;&nbsp;&nbsp;&nbsp;&nbsp;&nbsp;We provide examples of extracted masks during Masked Sampling for different sampling steps and binarization quantiles.

**Appendix F: Additional Examples** as requested by Reviewer JY4n

&nbsp;&nbsp;&nbsp;&nbsp;&nbsp;&nbsp;We provide more visual results for Sampling and baseline methods, complementing Figures 6 and 9 from the main text.

**Appendix G: Dreambooth results** as requested by Reviewers E8xN, 6r26, and WtXV

&nbsp;&nbsp;&nbsp;&nbsp;&nbsp;&nbsp;We perform a comparison of Sampling methods over the Dreambooth base method.

**Appendix H: Complex Prompts Setting** as requested by Reviewers E8xN and JY4n

&nbsp;&nbsp;&nbsp;&nbsp;&nbsp;&nbsp;We analyze the performance of Sampling methods and provide visual and numerical results using more complex prompts.

**Appendix I: DINO Image Similarity** as requested by Reviewers JY4n and WtXV

&nbsp;&nbsp;&nbsp;&nbsp;&nbsp;&nbsp;We compare CLIP Image Similarity with DINO Image Similarity as the main metric for estimating similarity between concept and generated images.

**Appendix J: PixArt-alpha & SD-XL** as requested by Reviewers E8xN and 6r26

&nbsp;&nbsp;&nbsp;&nbsp;&nbsp;&nbsp;We conduct experiments on more recent generative models and provide quantitative results similar to Figure 7.

**Appendix K: Updated Figure 1** as requested by Reviewer WtXV

&nbsp;&nbsp;&nbsp;&nbsp;&nbsp;&nbsp;We attach one more example of Mixed and Switching samplings where differences between sampling with concept and with superclass are more pronounced.


We believe we addressed all concerns presented during the rebuttal in the responses and updated manuscript. Given all the additional evidence and clarifications, we would appreciate reconsidering the current assessment of this work.

---

### Meta-Review · Area_Chair_hAxr · 2024-12-09

**Metareview:**

The paper receives mostly negative scores from reviewers. While reviewers appreciate the comprehensive comparisons between different sampling methods (JY4n), they found the results not surprising and questioned the generality to other models. The authors are encouraged to address these comments in the revised version.

**Additional Comments On Reviewer Discussion:**

During the discussion reviewer JY4n agrees the paper lacks deeper insights beyond simple comparisons between different sampling methods. The authors are encouraged to provide more insightful analysis or experiments to inspire the community more.

---

### Decision · Program_Chairs · 2025-01-22

Reject